# Histone methyltransferase activity affects metabolism in human cells independently of transcriptional regulation

**Marcos Francisco Perez**[1,2]*, **Peter Sarkies**[1]*

**1** Department of Biochemistry, University of Oxford, Oxford, United Kingdom, **2** Department of Cells and Tissues, Instituto de Biologia Molecular de Barcelona (IBMB), CSIC, Barcelona, Spain

* mpbbmc@ibmb.csic.es (MFP); peter.sarkies@bioch.ox.ac.uk (PS)

**Data Availability Statement:** All relevant data are contained within the paper and/or supporting information files. The data underlying the graphs shown in the figures and the code used can be found in https://zenodo.org/record/8383542.

## Abstract

The N-terminal tails of eukaryotic histones are frequently posttranslationally modified. The role of these modifications in transcriptional regulation is well-documented. However, the extent to which the enzymatic processes of histone posttranslational modification might affect metabolic regulation is less clear. Here, we investigated how histone methylation might affect metabolism using metabolomics, proteomics, and RNA-seq data from cancer cell lines, primary tumour samples and healthy tissue samples. In cancer, the expression of histone methyltransferases (HMTs) was inversely correlated to the activity of NNMT, an enzyme previously characterised as a methyl sink that disposes of excess methyl groups carried by the universal methyl donor S-adenosyl methionine (SAM or AdoMet). In healthy tissues, histone methylation was inversely correlated to the levels of an alternative methyl sink, PEMT. These associations affected the levels of multiple histone marks on chromatin genome-wide but had no detectable impact on transcriptional regulation. We show that HMTs with a variety of different associations to transcription are co-regulated by the Retinoblastoma (Rb) tumour suppressor in human cells. Rb-mutant cancers show increased total HMT activity and down-regulation of *NNMT*. Together, our results suggest that the total activity of HMTs affects SAM metabolism, independent of transcriptional regulation.

## Introduction

The discovery of strong associations between the transcriptional states of genes and methylation of histones at these loci was a seminal moment in the study of gene regulation [1]. The concept of the histone code [2] proposed that knowledge of the particular combination of epigenetic marks in chromatin would allow a deterministic prediction of gene expression, just as the deciphering of the genetic code had allowed precise prediction of gene products. However, 2 decades later, despite strong correlations of some histone marks with specific transcriptional states [3], these associations can be ambiguous [4,5], while evidence of causal links between histone marks and transcriptional activation or repression remains equivocal [6].

In that time, it has become widely appreciated that histone modifications can be influenced by cellular metabolism [7]. Histone methylation is influenced by the availability of S-adenosyl methionine (SAM, also abbreviated as AdoMet or SAMe), the universal methyl donor that is

**Funding:** This work was supported by the UAS John Fell Fund (to PS) grant number 0011417, which supported the salary MFP until January 2023. This work was also funded by a Ramon y Cajal fellowship RYC2021-034496-I (to MFP), which supported the salary of MFP from January 2023. The funders had no role in study design, data collection and analysis, decision to publish, or preparation of the manuscript.

**Competing interests:** The authors have declared that no competing interests exist.

**Abbreviations:** 1MNA, 1-methylnicotinamide; BRCA, breast cancer; CCLE, Cancer Cell Line Encyclopedia; COAD, colon adenocarcinoma; FDR, false discovery rate; GRN, gene regulatory network; GTEx, Genotype-Tissue Expression; HERV, human endogenous retrovirus; HMT, histone methyltransferase; LC–MS, liquid chromatography–mass spectrometry; LUSC, lung squamous cell carcinoma; MEF, mouse embryonic fibroblast; MRN, median-ratio normalisation; NNMT, nicotinamide N-methyltransferase; PAAD, pancreatic adenocarcinoma; PC, phosphatidylcholine; PE, phosphoethanolamine; PI, proliferative index; PRMT, protein arginine methyltransferase; Rb, Retinoblastoma; SAH, S-adenosyl homocysteine; SAM, S-adenosyl methionine; STAD, stomach adenocarcinoma; TCGA, The Cancer Genome Atlas; TR, transcriptional regulator; TSS, transcription start site.

required for cellular methylation of lipids, proteins, nucleic acids, and metabolites, and which can be modulated by dietary methionine supplementation [8]. However, the abundance of histones in the cell offers the potential for histone modifications to impact metabolism [9]. Histones have the potential to act as a methyl sink, as histone methylation consumes methyl groups from SAM but demethylation releases formaldehyde, which cannot be easily recycled to release methyl groups [10]. Methyl sinks play key metabolic roles, acting to buffer the ratio of SAM to S-adenosyl homocysteine (SAH, also abbreviated as AdoHcy) and supporting the synthesis of important sulphur-containing metabolites such as cysteine and glutathione via the SAH-dependent transsulphuration pathway [9,11].

Here, we discovered strong negative relationships between the total expression of histone methyltransferases (HMTs) and metabolic pathways previously characterised as methyl sinks. In both cancer cells and healthy tissue, we show that these relationships affected genome-wide levels of histone posttranslational modifications but did not have significant consequences for transcriptional regulation. We show that HMTs were co-expressed and negatively regulated by the Retinoblastoma (Rb) tumour suppressor in cancer. Our data suggest the hypothesis that the total activity of HMTs has consequences for SAM homeostasis in healthy human tissues and tumours, independent of the functions of histone methylation in transcriptional regulation.

## Results

### Histone methyltransferase expression correlates to cellular metabolite levels

We set out to investigate a potential link between metabolism and histone methylation. We reasoned that effects of HMT activity on metabolism might result in correlations between HMT levels and cellular metabolite concentrations. To investigate this possibility, we used a publicly available metabolomics dataset consisting of 225 metabolites profiled by liquid chromatography–mass spectrometry (LC–MS) across 911 cell lines from the Cancer Cell Line Encyclopedia (CCLE), representing more 23 cancer types [12]. We related metabolite levels to the normalised expression of HMTs in the same cell line. We curated a list of 38 HMTs (S1 Table) and examined the correlation of each HMT to all metabolites. Across this set 1-methylnicotinamide (1MNA) consistently emerged as the metabolite most strongly associated to HMTs, with a false discovery rate (FDR) < 0.05 for 18 individual HMTs, a geometric mean FDR of 0.001 (S2 Table) and an average Pearson's correlation of −0.090 (range −0.261 to 0.160). Indeed, 1MNA was the metabolite with the largest absolute correlation to the total level of HMTs obtained by summing the expression of the 38 individual enzymes (Fig 1A; Pearson's correlation = −0.274, FDR = $1.77 \times 10^{-14}$).

### HMT expression varies reciprocally with the activity of the 1MNA/NNMT methyl sink

The strong relationship between 1MNA and HMT levels indicated that HMT expression variation across cancer cell lines might be associated with changes in metabolism. To investigate which metabolic pathways might be responsible, we performed principal component analysis (PCA) and clustering analysis on all 225 metabolites. Related metabolites from known biochemical pathways tended to cluster together. 1MNA was a clear outlier in both analyses (Figs 1C and S1), indicating that 1MNA synthesis reflects a discrete metabolic process. Indeed, 1MNA is known to be a stable metabolic end-product that has no downstream metabolites in cancer [13] and is excreted from cells in healthy tissues [14].

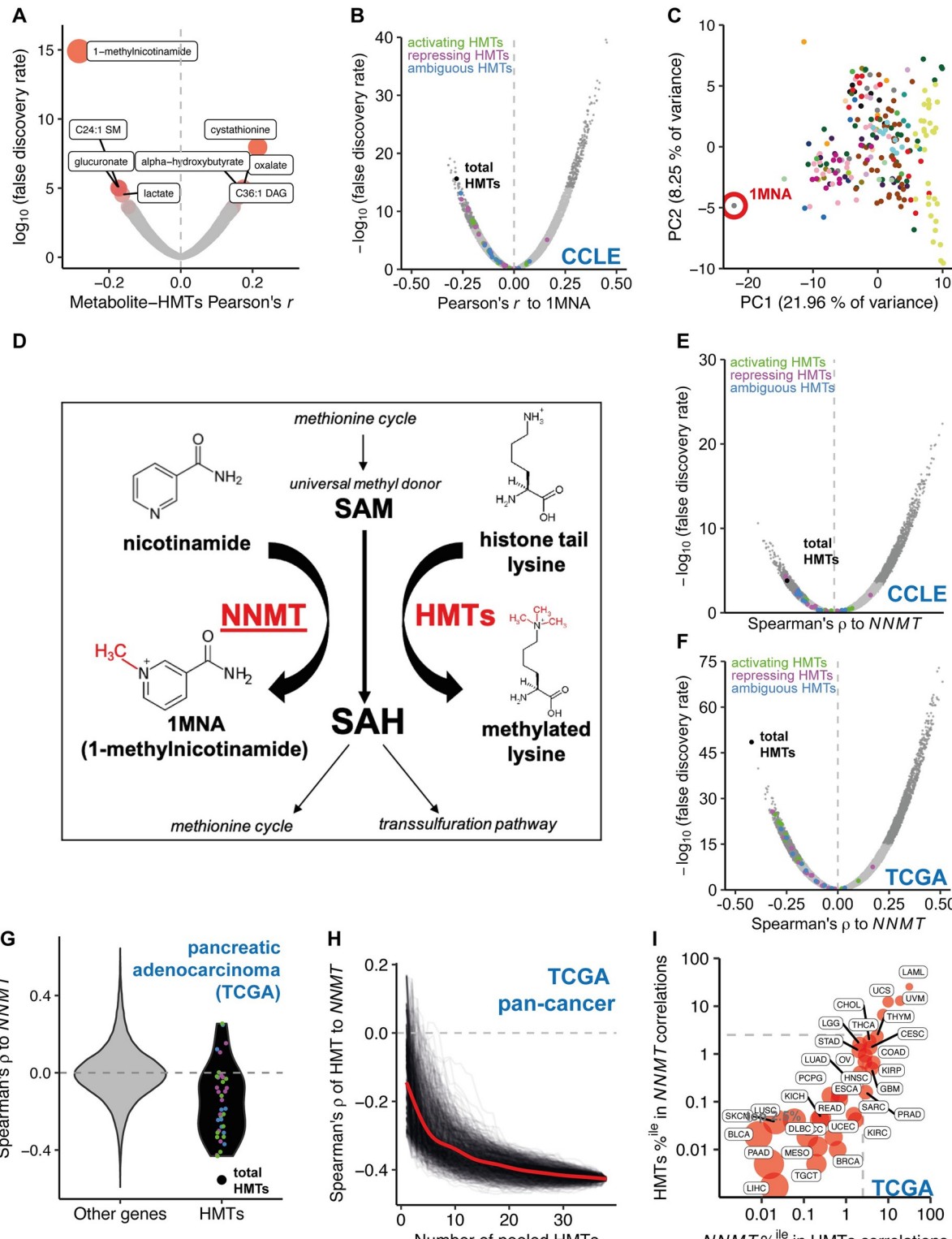

**Fig 1. Total HMT expression is strongly anticorrelated with the activity of NNMT in cancers.** (A) Volcano plot showing Pearson's correlation and FDR for 225 metabolites to total HMT expression (total RNA-seq median of ratios-normalised pseudocounts) across 927 cancer cell lines from the CCLE. (B) Volcano plot showing Pearson's correlation and FDR for expression of 10,275 expressed genes to levels of 1MNA across 927 CCLE cancer cell lines. The top and bottom 2.5% of points are shown in darker grey. HMT-encoding genes are shown as points coloured according to their association with transcriptional activation (green), repression (magenta), or an unclear relationship

(blue). Pearson's r for total HMT expression is shown as a black point. (C) PCA of metabolite levels across 927 cancer cell lines from the CCLE. 1MNA is highlighted with a red circle. (D) NNMT and HMTs both convert SAM to SAH and so can affect cellular methylation potential by acting as a "sink." (E) Volcano plot showing Spearman's correlation and FDR for expression of *NNMT* vs. 52,440 genes in a pan-cancer analysis of 927 CCLE cell lines across 23 cancer types. HMT-encoding genes are shown as points as in panel 1B. (F) Volcano plot showing Spearman's correlation and FDR for expression of *NNMT* vs. 60,489 genes in a pan-cancer analysis of TCGA primary tumours across 33 cancer types. HMT-encoding genes are shown as points as in panel 1B. (G) Violin plot showing Spearman's correlation to *NNMT* for HMTs (black, right) or other genes (left, grey) in 79 primary ACC tumours from the TCGA. Individual HMT-encoding genes are shown as points as in panel 1B. (H) Spearman's correlation vs. *NNMT* expression of total expression of pooled HMTs added to the pool in a random order, and 1,000 individual iterations are shown as black lines, with the locally estimated smoothing (Loess fit) trendline shown in red. (I) TCGA pan-cancer analysis showing rank percentile position of total HMTs among correlations of *NNMT* expression to 60,489 genes and vice versa in 33 distinct cancer types. Bubble size is inversely proportional to the log of the "relative reciprocal score," the sum of squares of the ranks of total HMTs/*NNMT* in the reciprocal distribution. The dashed grey box indicates correlations in the strongest 2.5% of anticorrelated genes. Underlying data for all panels can be found in https://zenodo.org/record/8383542. 1MNA, 1-methylnicotinamide; ACC, adrenocortical carcinoma; CCLE, Cancer Cell Line Encyclopedia; FDR, false discovery rate; HMT, histone methyltransferase; NNMT, nicotinamide N-methyltransferase; SAH, S-adenosyl homocysteine; SAM, S-adenosyl methionine; TCGA, The Cancer Genome Atlas.

1MNA is the product of methylation of nicotinamide by the enzyme nicotinamide N-methyltransferase (NNMT). 1MNA levels were strongly correlated with *NNMT* expression as measured by RNA-seq (S2A Fig and S2 Table) and with NNMT protein levels as measured by quantitative mass spectrometry (S2B Fig and S3 Table; [15]), while nicotinamide levels were anticorrelated with *NNMT* (S2 and S3 Tables). We conclude that 1MNA levels reflect the activity of the metabolic pathway that converts nicotinamide to 1MNA, catalysed by the enzyme NNMT. Moreover, this pathway is not tightly coupled to the activity of other pathways of core metabolism. We therefore decided to investigate possible explanations for the relationship between HMT levels and 1MNA pathway activity.

## HMT and NNMT are alternative pathways that both consume methyl groups

The reaction catalysed by NNMT uses SAM as a cofactor, transferring a methyl group from SAM to nicotinamide to form 1MNA (Fig 1D). NNMT has been proposed to function as a "sink" for methyl groups. High NNMT activity can reduce the SAM:SAH ratio [13,16–18].

To further investigate the relationship between the 1MNA synthesis pathway and HMT activity, we investigated the relationship between *NNMT* expression and total HMT expression. NNMT protein levels in 2 cancer cell line panels, the CCLE and NCI60 [19], correlated strongly to *NNM*T expression (S2C and S2D Fig, respectively). *NNMT* expression is therefore a reliable indicator of NNMT protein levels and catalytic activity. *NNMT* expression and total HMT expression in the CCLE were negatively correlated (Fig 1E). HMT protein levels were negatively correlated with NNMT protein levels both for individual HMTs (for 20 HMTs detected in >90% of samples, mean Pearson's *r* to NNMT protein levels = −0.108, range −-0.306–0.199, FDR < 0.05 negative correlation for 12/20 and positive correlation for 1/20) and collectively (S2E Fig; mean Pearson's *r* with sample mean HMT protein Z-score = −0.244, *p*-value = $2.62 \times 10^{-5}$). HMT protein levels are also negatively correlated with 1MNA levels (S3 Table; mean Pearson's *r* with sample mean HMT protein Z-score = −0.102, *p*-value = $7.94 \times 10^{-2}$). Altogether, this suggested that elevated 1MNA synthesis is associated with reduced HMT activity.

One possible explanation for the negative association between HMT levels and 1MNA synthesis is that 1MNA directly represses HMT transcription. However, partial correlation analysis indicated that the correlation between HMTs and 1MNA was weakened from −0.280 to −0.073 after controlling for *NNMT* expression (S2F Fig). We concluded that 1MNA itself is unlikely to regulate HMT expression. Instead, HMT levels were primarily associated with *NNMT* expression, and thus with the rate of 1MNA synthesis, rather than 1MNA levels

themselves. These findings are consistent with the proposal that HMT activity and NNMT activity are parallel pathways capable of acting as methyl sinks [9]. NNMT activity is reduced when HMT activity is high (Fig 1E)

## HMT and *NNMT* expression are tightly coupled in primary tumours

To test if the relationship we uncovered in cancer cell lines was also seen in tumours, we used RNA-seq data from primary tumours from 33 distinct cancer types found in the Cancer Genome Atlas (TCGA) database to interrogate the correlation between HMTs and *NNMT* expression. The HMT-*NNMT* expression relationship was much stronger than that observed in the CCLE cell lines, both in a pan-cancer analysis (Fig 1E and 1F) and within individual cancer types (S1 and S2 Files). The relationship was strengthened by pooling the expression of HMTs together (Fig 1F and S1 and S2 Files). As an example, among the 177 pancreatic adenocarcinoma (PAAD) primary tumour samples, the relationship of *NNMT* expression with expression of individual HMTs was for the most part not exceptionally strong. However, when their expression was pooled, HMT expression anticorrelated with *NNMT* better than almost any single gene (rank 3, top 0.00496%; Fig 1G). Conversely, *NNMT* was one of the most negatively correlated genes to pooled HMT expression (rank 9, top 0.0149% of genes overall).

Raw correlation statistics (such as $\rho$) do not necessarily provide a reliable comparator of the strength of the relationship, as the distribution of gene correlations can differ between cancer or tissue types [20]. To overcome this, we computed a relative reciprocal relationship score for HMTs and *NNMT* (see Methods). The relative reciprocal relationship scores were much stronger in the TCGA primary tumours than in the CCLE cell lines (Figs 1I and S2G). The HMT-*NNMT* relationship was found across most cancer types and was particularly strong in liver, pancreatic, and bladder cancers. The only cancer in which this relationship was not evident at all was acute myeloid leukemia (LAML). This may be a result of the low expression of *NNMT* in this cancer type (S2H Fig). Indeed, across all cancers there was a significant correlation between the strength of the relationship between *NNMT* and HMT levels and the expression of *NNMT* (S2I Fig). Importantly, the HMT-*NNMT* relationship remained robust when controlling for immune cell infiltration, as estimated by 2 distinct gene expression deconvolution tools [21,22] (S3 Fig).

To visualise how the correlation strengthened as HMT expression was combined, we performed a pan-cancer analysis in which we added successively more HMTs and calculated the correlation between these HMTs and *NNMT*. The relationship between HMTs and *NNMT* became stronger as more HMTs were added, regardless of the order with which the HMTs were combined (Fig 1H). This relationship indicates that the anticorrelation between HMTs and *NNMT* is distributed across HMTs and not due to 1 or 2 HMTs with unusually strong anticorrelations. Using a simple stochastic modelling approach, we determined that the reinforcement of this relationship when more HMTs are added is consistent with co-regulation of the HMTs at the transcriptional level (S2J Fig).

Total HMT expression was more strongly anticorrelated to *NNMT* than any individual HMT in 16/33 cancer types (S2 File). However, 1 notable exception was melanoma (SKCM), which despite being one of the cancer types with the strongest *NNMT*-total HMT relationships exhibited a far stronger anticorrelation of *NNMT* to the euchromatic H3K9me3 writer *SETDB1* (S4 Fig and S2 File). *SETDB1* is the most strongly anticorrelated gene in the genome to *NNMT* and vice versa. As *SETDB1* is a recurrently amplified, established driver of melanoma [23–26], this raises the possibility that increased *SETDB1* activity might have important metabolic consequences for melanoma.

## Among cellular methyltransferases, HMTs have the strongest relationship with *NNMT*

To test whether the strong anticorrelation to *NNMT* is specific to histone lysine methyltransferases, we calculated the relative reciprocal relationship scores for protein arginine methyltransferases (PRMTs), DNA methyltransferases, and groups of RNA methyltransferases (S5 Fig; gene sets in S1 Table). The relationship of *NNMT* to histone lysine methyltransferases was the strongest and most widespread. There was no significant relationship between *NNMT* and PRMTs or tRNA/rRNA methyltransferases. In breast cancer (BRCA) and 16 other cancer types *NNMT* had a strong negative relationship to mRNA methyltransferases. In lung squamous cell carcinoma (LUSC) and 4 other cancer types, there was a strong negative relationship between *NNMT* and small RNA methyltransferases. More notably, 9 out of 33 cancer types displayed a strong relationship between *NNMT* and methyltransferases with unknown substrates, with very strong relationships evident in LUSC and stomach and colon adenocarcinomas (STAD and COAD). We suggest that other cellular methyltransferases may also act as methyl sinks in parallel to lysine HMTs and *NNMT*. However, this relationship is most evident for histone lysine methyltransferases across cancer.

## The relationship between HMTs and NNMT is specific to cancer

We investigated the relationship of HMT to *NNMT* in RNA-seq data from healthy tissue samples available through the Genotype-Tissue Expression (GTEx) project. A strong relationship between *NNMT* expression and HMT expression was observed in only 1 individual tissue type (muscle) (Fig 2A and 2B and S3 File). Moreover, in 10 out of the 12 TCGA cancer types for which matched tumour and normal tissue samples from at least 30 patients were available, the negative relationship of *NNMT* and HMT was stronger in the cancer samples relative to the matched normal samples (paired Wilcoxon test on reciprocal scores, *p*-value = $1.47 \times 10^{-3}$; S6A Fig).

We wondered whether the difference between cancer and healthy tissue could be due to differences in the expression levels of *NNMT* and HMTs. However, while *NNMT* has frequently been reported as overexpressed in cancers [27–34], we did not find strong support for this across the TCGA. *NNMT* was significantly up-regulated in tumour samples compared to matched normal samples in 3/12 cancer types, with 4/12 showing significant down-regulation (S6B Fig). Similarly, total HMT expression was not consistently altered in cancer, being up-regulated in 4/12 cancer types and down-regulated in 2/12 (S6C Fig). Interestingly, however, we observed that these changes were inversely correlated: cancers which up-regulate HMTs have reduced *NNMT* expression and vice versa (S6D Fig).

## In healthy tissues, HMTs correlate with *PEMT*, an alternative methyl sink

We tested whether total HMT expression might show an analogous relationship to another methyltransferase or class of methyltransferases operating within the cell. In 18/48 tissues total HMT expression anticorrelated strongly with *PEMT* (reciprocally in top 2.5% of genes; Fig 2C and S4 File). Interestingly, a particularly high proportion of brain tissues (10/13) showed a strong, significant relationship. Across the 18 tissues that showed strong associations between *PEMT* and HMT expression (Fig 2D), the relationship became stronger as more HMTs were pooled (Fig 2E and 2F). *PEMT* was strongly negatively correlated to HMTs in 7/33 cancers (Fig 2G).

PEMT is an enzyme that adds 3 methyl groups to the phospholipid phosphoethanolamine (PE) to produce phosphatidylcholine (PC) (Fig 2H). PC makes up around 40% of the lipid

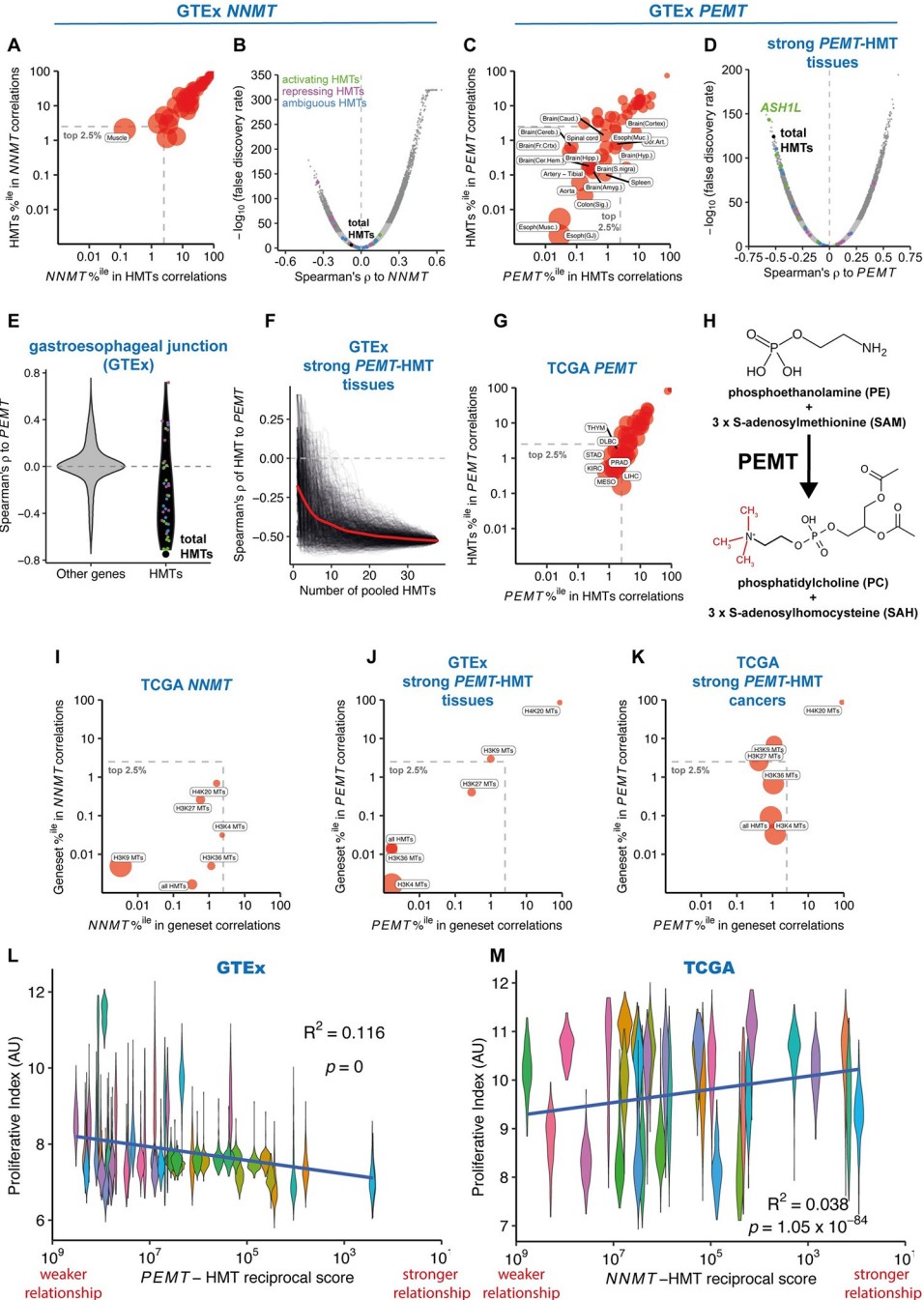

**Fig 2. Total HMT expression is strongly anticorrelated with the expression of *PEMT* in healthy tissues.** (A) Analysis showing rank percentile position of total HMTs among correlations of *NNMT* expression to 56,200 genes and vice versa in 48 distinct healthy tissue types from the GTEx project. Bubble size is inversely proportional to the log of the "relative reciprocal score," the sum of squares of the ranks of total HMTs/*NNMT* in the reciprocal distribution (see Methods). The dashed grey box indicates correlations in the strongest 2.5% of anticorrelated genes, with tissues labelled. (B) Volcano plot showing Spearman's correlation and FDR for expression of *NNMT* vs. 56,200 genes in a pan-cancer analysis of GTEx primary tumours across 48 tissue types. HMT-encoding genes are shown as points coloured according to association with transcriptional regulation; correlation for total HMT expression is shown as a black point. (C) Analysis showing rank percentile position of total HMTs among correlations of *PEMT* expression and vice versa in healthy tissue types from the GTEx project. Bubble size and dashed grey box as in panel 2A. (D) Volcano plot showing Spearman's correlation and FDR for expression of *PEMT* vs. 56,200 genes in a cross-tissue analysis of 18 tissue types with a strong HMT-*PEMT* relationship (within the grey box in panel 2C). HMT-encoding genes are shown as points as in panel 2B. (E) Violin plot showing Spearman's correlation to *PEMT* of HMTs (black, right) or other

genes (left, grey) in 375 patient samples from the gastroesophageal junction. HMT-encoding genes are shown as points as in panel 2B. (F) Spearman's correlation vs. *PEMT* expression of total expression of pooled HMTs added to the pool in a random order in a cross-tissue analysis of tissues with a strong HMT-*PEMT* relationship (within the grey box in panel 2C); 1,000 individual iterations are shown as black lines, with Loess fit trendline in red. (G) Analysis showing rank percentile position of total HMTs among correlations of *PEMT* expression to 60,489 genes and vice versa in 33 cancer types from the TCGA. Bubble size and dashed grey box as in panel 2A. (H) PEMT sequentially methylates phosphoethanolamine to produce PC, converting 3 molecules of SAM to SAH. (I) Analysis showing rank percentile position of HMTs classified by their substrate histone lysine residues among correlations of *NNMT* expression and vice versa in cancer types from the TCGA. Bubble size and dashed grey box as in panel 2A. (J) Analysis showing rank percentile position of HMT sets methylating distinct histone lysine residues among correlations of *PEMT* expression to 56,200 genes and vice versa in a pan-tissue analysis of 18 tissue types from the GTEx with a strong HMT-*PEMT* relationship (within the grey box in panel 2C). Bubble size and dashed grey box as in panel 2A. (K) Analysis showing rank percentile position of HMT sets methylating distinct histone lysine residues among correlations of *PEMT* expression and vice versa in a pan-cancer analysis of 7 cancer types from the TCGA with a strong HMT-*PEMT* relationship (within the grey box in panel 2G). Bubble size and dashed grey box as in panel 2J. (L) Violin plot showing healthy tissue sample PI, a measure of proliferation inferred from sample RNA-seq gene expression data, for 48 tissue types of the GTEx arranged by the strength of the anticorrelating relationship between *PEMT* and total HMTs. Note the x axis is inverted as a lower relative reciprocal score indicates a stronger relationship. (M) Violin plot showing tumour PI for 31 cancer types of the TCGA arranged by the strength of the anticorrelating relationship between *NNMT* and total HMTs. Underlying data for all panels can be found in https://zenodo.org/record/8383542. FDR, false discovery rate; GTEx, Genotype-Tissue Expression; HMT, histone methyltransferase; NNMT, nicotinamide N-methyltransferase; PC, phosphatidylcholine; PI, proliferative index; SAH, S-adenosyl homocysteine; SAM, S-adenosyl methionine; TCGA, The Cancer Genome Atlas.

content of the plasma membrane in eukaryotic cells [35]. PEMT activity contributes around 30% of cellular PC synthesis [36]. The abundance of PC in the membrane suggested the possibility that PEMT could act as a sink for methyl groups, similarly to NNMT; indeed, PEMT has been suggested to be the primary consumer of SAM in mammals [37].

We investigated whether other groups of methyltransferases also anticorrelated to *PEMT*. We found a significant negative relationship of *PEMT* to mRNA methyltransferases in 14 tissue types, including 9/13 brain tissues (S7 Fig). Likewise, 7 tissues (5 from the brain) displayed an analogous relationship with DNA methyltransferases (S7 Fig). Histone lysine methyltransferases had the strongest and most consistent anticorrelation to *PEMT*.

## Differential contributions of methylated residues to HMT relationship with methyl sinks

We investigated whether specific methylated residues on histones contribute more strongly to the relationship with cellular methyl sinks. In cancer, the HMTs that methylate the H3K9 residue showed a stronger relationship to *NNMT*, while the relationship to methyltransferases targeting other residues was weaker, albeit still highly significant (Fig 2I). In healthy tissues *PEMT* showed a strong reciprocal relationship to H3K4 and H3K36 methyltransferases, whereas the relationship with H3K9 and H3K27 methyltransferases, associated with transcriptional repression, was weaker and no strong relationship existed for H4K20 methyltransferases (Fig 2J). The difference between PEMT and NNMT was not due to a difference between cancer and healthy tissue because cancers with a significant HMT-*PEMT* relationship showed similar residue specificity as healthy tissues (Fig 2K).

We wondered whether the difference in HMTs that correlated with *NNMT* and *PEMT* could be due to the cell cycle activity of these pathways. Transcription-associated methylation on H3K4me3 and H3K36me3 is enriched in quiescent cells relative to methylation of H3K9me3, which largely occurs in late S and G2 phase to restore H3K9 methylation to newly synthesised histones [38]. *PEMT* expression is also reported to vary strongly across the cell cycle, peaking in G1 phase and declining in S phase [39,40]. We tested the correlation between the strength of HMT-*PEMT* relationship across the GTEx healthy tissues to the tissue sample's

proliferative index (PI), a measure of proliferation inferred from RNA-seq data [41]. The HMT-*PEMT* relationship tended to be stronger in tissues with a lower PI (Fig 2L; $R^2$ = 0.116, *p*-value ~ 0). This may reflect the inability of proliferating cells to use PEMT as a methyl sink throughout the cell cycle. The HMT-*NNMT* relationship in cancers exhibited the opposite trend, albeit weakly (Fig 2M; $R^2$ = 0.0384, p-value = $1.05 \times 10^{-84}$). The aggressive proliferation of cancer cells may explain why total HMT expression correlates to different methyl sinks in cancers and healthy tissues.

## Histone methylation in chromatin is correlated to methyl sink activity without changes in transcription

We investigated whether the relationship between HMT activity and *PEMT* or *NNMT* impacted histone methylation levels in chromatin genome-wide. We used ChIP-seq data from healthy tissues and cancer cell lines to assess histone methylation levels. In healthy tissues, we found that there was a global negative relationship of PEMT expression with H3K4me3, H3K9me3, and H3K27me3 at all classes of genomic regions examined. For example, 99% of 12,355 gene promoters marked by H3K4me3 peaks showed an anticorrelated relationship between *PEMT* expression and total H3K4me3 signal or H3K4me3 peak width, an orthogonal measure of histone methylation levels (Figs 3A, 3B, S8A and S8B). Similarly, 99% of 2,870 repetitive regions modelled showed an anticorrelated relationship for H3K9me3 signal and *PEMT* expression (Fig 3A). However, this was not the case for H3K36me3, which had a moderately positive relationship with *PEMT* expression at promoters and gene bodies. No negative relationships with *NNMT* expression were observed in healthy tissues (S8C Fig).

Altered histone modification levels are often associated with changes in transcription of the genes at the corresponding loci. However, we found that despite low signal of H3K4me3 (associated with transcription) and H3K9me3 (associated with repression) in high-*PEMT* samples, expression from marked genes was not affected in either case (S8D Fig). Similarly, we found that variation across samples in total H3K4me3 signal (S8E Fig) or width (S8F Fig) at marked promoters, and H3K9me3 signal on H3K9me3-marked gene bodies, does not correlate with the expression of the corresponding genes. Thus in healthy tissues, *PEMT* varied with histone methylation levels independent of effects on transcription.

In cancer, H3K9me3 and H4K20me3 at both gene bodies and promoters were anticorrelated to cell line *NNMT* expression as measured by RNA-seq (Fig 3C), microarrays (S9A Fig), and proteomics (S9B Fig); however, H3K4me3 showed a positive relationship. No negative relationships were observed for *PEMT* expression (S9C Fig). Similarly, we found that both H3K9me3 and H4K20me3 levels at multiple classes of repetitive elements were negatively correlated with *NNMT* expression (Fig 3D). For all classes, the negative correlation was stronger at genomic sites with a higher average ChIP-seq signal across samples (S9D Fig). The anticorrelation with *NNMT* expression was particularly strong at centromeric satellites (Fig 3D), independent of their tendency to display higher signal of heterochromatic marks than other classes of repetitive element (S9E Fig).

While variation across samples in H4K20me3 signal at H4K20me3-marked gene bodies was negatively correlated with *NNMT* levels, expression from those genes displayed little relationship with *NNMT* (S9F Fig). We also estimated locus-specific expression of transposable elements, specifically human endogenous retroviruses (HERVs). Similarly, we found that despite reduced signal of this canonically repressive histone mark at HERVs in samples with high *NNMT* expression, HERV expression was not increased (S9F Fig). Indeed, variation in total signal of H3K9me3/H4K20me3 at marked sites was not associated with the level of transcription from either gene bodies or HERVs (S9G Fig).

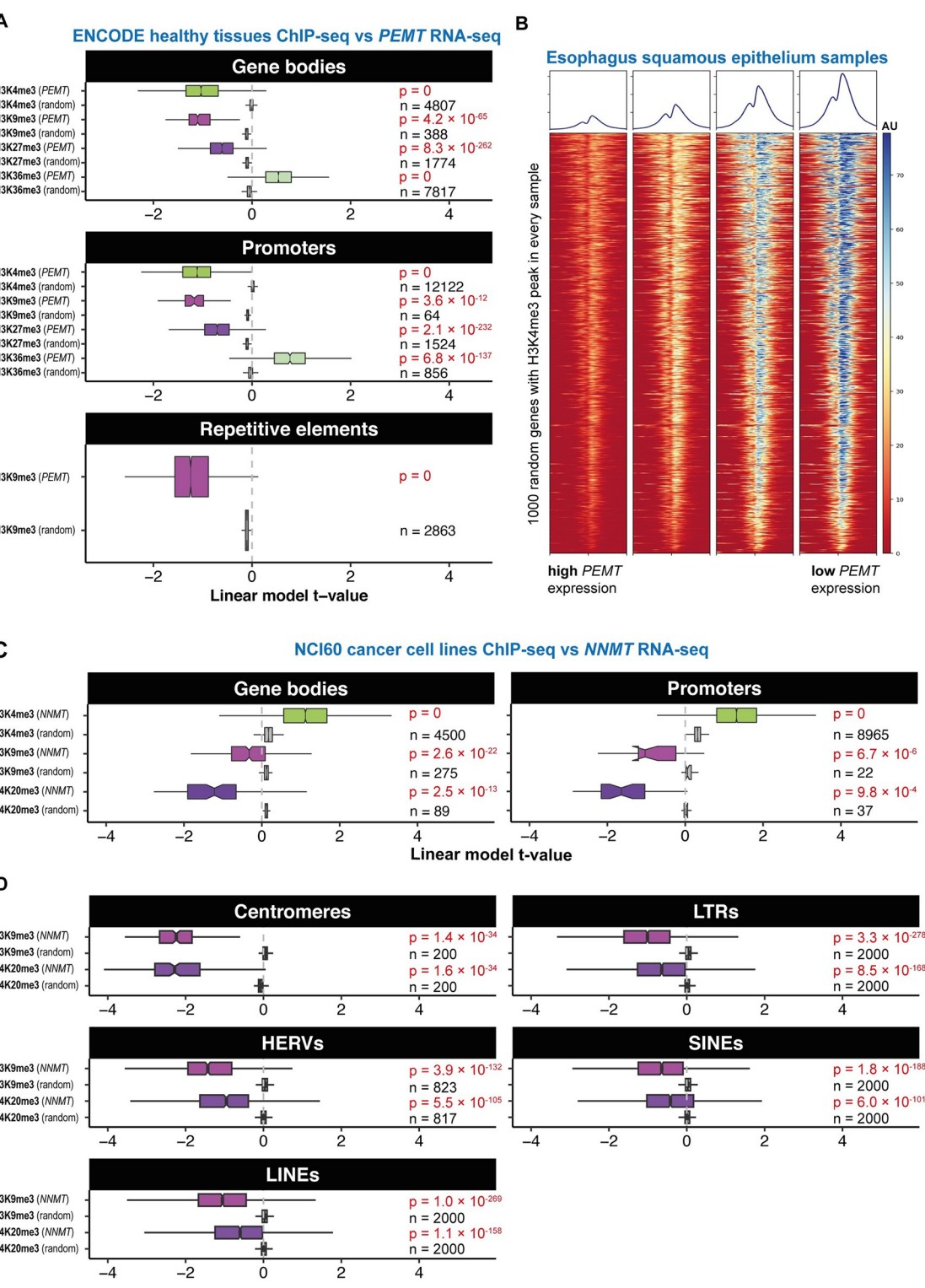

**Fig 3. *PEMT* and *NNMT* expression anticorrelate globally with levels of specific histone marks genome-wide in healthy tissues and cancers, respectively.** (A) Boxplot shows t-values from linear mixed effects model for sample *PEMT* expression predicting ChIP-seq signal for various histone marks (label left) on gene bodies, promoters or repetitive elements (subpanel headers) in patient tissue samples collected as part of the ENCODE project. The number of individual sites is noted on the plot for each boxplot; *p*-values derive from paired Wilcoxon tests against a null distribution calculated by the mean t-value at each locus for 1,000 random

expressed genes. (B) Heatmap showing H3K4me3 ChIP-seq signal (log$_2$ fold change over input) over 1,000 random genes for 4 samples from the squamous epithelium of the esophagus arranged in order of *PEMT* expression. (C) Boxplot shows t-values from generalised linear models for *NNMT* expression (RNA-seq) predicting ChIP-seq signal for various histone marks (label left) on gene bodies and promoters in cell lines of the NCI60 cancer cell line panel. The number of individual sites is noted on the plot for each boxplot; *p*-values derive from paired Wilcoxon tests against a null distribution calculated by the mean t-value at each locus for 1,000 random expressed genes. (D) Boxplot shows t-values from generalised linear models for *NNMT* expression (RNA-seq) predicting ChIP-seq signal for various histone marks (label left) on different classes of repetitive elements in cell lines of the NCI60 cancer cell line panel. The number of individual sites is noted on the plot for each boxplot; *p*-values derive from paired Wilcoxon tests against a null distribution calculated by the mean t-value at each locus for 1,000 random expressed genes. Sites shown are from bin with highest ChIP signal (cf. S9D Fig). Underlying data for all panels can be found in https://zenodo.org/record/8383542. HERVs, human endogenous retroviruses; LINEs, long interspersed nuclear elements; LTRs, long terminal repeats; NNMT, nicotinamide N-methyltransferase; SINEs, short interspersed nuclear elements.

## Histone methyltransferase genes are co-expressed

We wished to understand the structure and origin of the variation we observe in total HMT expression. We found that the expression of HMT genes was significantly more positively correlated to each other than to random genes (S10A and S10B Fig). The strongest co-expression was evident among the most highly expressed HMTs, which had a range of target lysine residues with divergent associations with transcription (see S10A Fig sidebars). This core of 14 to 16 highly expressed and highly correlated HMT genes was largely stable between healthy tissues and cancer (S10A Fig) and across individual tissue or cancer types (S5 and S6 Files). We performed gene co-expression network analysis, showing a similar network architecture with a small number of modularity classes in both cancer and healthy tissue (S10C and S10D Fig). The strength of network edges was highly concordant between pan-cancer and pan-tissue correlation analyses (S10C and S10D Fig; Pearson's correlation = 0.834; Jaccard index on network with |edge strength| > 0.2 = 0.569). While distinct modularity clusters showed some similarity to annotated associations with transcription (S10C and S10D Fig), the overlap was not strong. Together, this suggested the possibility that HMTs might be coordinately regulated independent of their transcriptional functions.

## Histone methyltransferase genes are regulated by E2F and Rb

Seeking to understand the possible basis for co-regulation of HMTs, we turned to *Caenorhabditis elegans*, a model organism with a simpler genetic regulatory architecture [42]. We observed a positive correlation among HMTs across 206 diverse natural genetic backgrounds in *C. elegans*, with the strongest correlation in a core of 10 genes largely consisting of the most highly expressed HMTs (S11A Fig). Genes in this cluster were more likely to have orthologues in the human highly expressed cluster (S11A Fig; odds ratio = 11.07, Fisher's exact test *p* = 0.0108). Additionally, we observed very strong negative correlations between total HMT expression and expression of the *NNMT* orthologues *anmt-1/3* and the *PEMT* analogue, *pmt-1* (S11B Fig). This relationship was partially due to varying levels of *anmt-1/3* and *pmt-1* across development (S11C Fig) and was also evident when controlling for developmental age (S11D and S11E Fig).

We performed a de novo motif enrichment search on the upstream regions of co-regulated *C. elegans* HMT genes. The most strongly enriched motif, present in 9/10 genes, resembled the binding motif of the E2F orthologue EFL-2 (Fig 4A). E2F transcription factors can be bound by the Rb protein, which represses transcription of E2F targets [43]. Using previously published genome-wide ChIP-seq data [44], we observed an enrichment for binding of LIN-35, the *C. elegans* Rb orthologue, close to the transcription start site (TSS) of HMT genes in the highly expressed cluster relative to other HMTs or to random genes (Fig 4B). Indeed, in RNA-seq data from *lin-35* mutants [45,46], we saw total HMT expression increased by 12% to 15% (1-way ANOVA, *p* = 0.053).

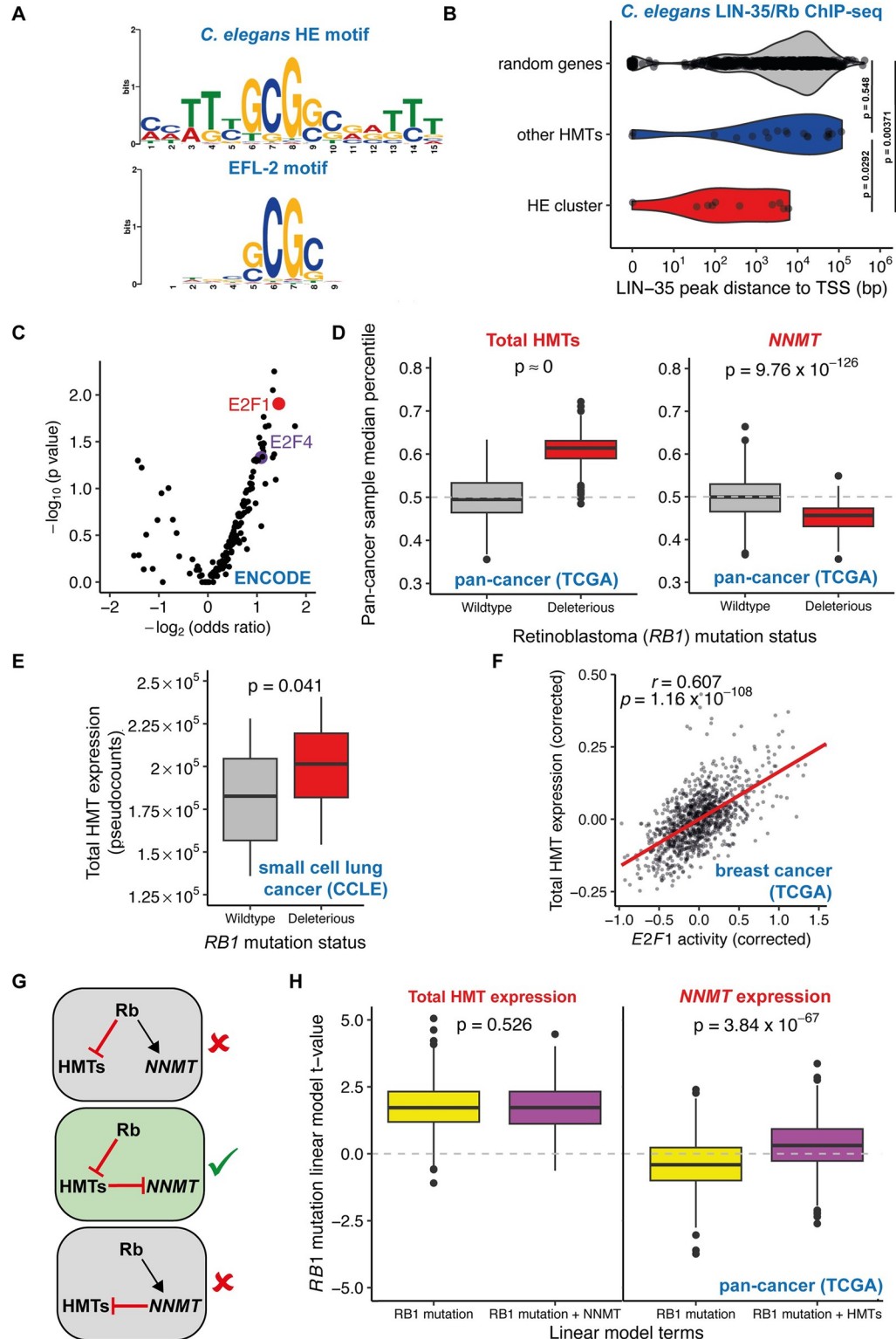

**Fig 4. HMTs are regulated by E2F and Retinoblastoma, with *NNMT* expression reduced downstream of HMTs in Rb-mutant cancers.** (A) Above: Sequence motif enriched in *C. elegans* HE cluster HMT promoters, relative to other HMTs promoters. Below: previously reported EFL-2 binding motif. (B) Binding of the *C. elegans* Retinoblastoma orthologue LIN-35 upstream of the TSS of the HE cluster, other HMT genes, and random genes; *p*-values from Wilcoxon test. (C) Enrichment for transcription factor binding, from ENCODE ChIP-seq experiments, upstream of

human HMT genes. Odds ratios and *p*-value derived from Fisher's exact test. (D) Boxplots show median total HMT or *NNMT* expression percentile drawn from 1,000 iterations of pan-cancer sampling of tumours with wild-type *RB1* or potentially deleterious *RB1* mutations; *p*-value derived from *t* test. (E) Total HMT expression in small cell lung cancer cell lines from the CCLE with wild-type *RB1* or deleterious *RB1* mutations; *p*-value derived from *t* test. (F) Estimated E2F1 activity vs. total HMT expression (both corrected for confounders) in breast cancer primary tumours from the TCGA. (G) Potential architectures of the GRN linking *RB1*, *NNMT*, and HMTs. (H) Linear model t-values explaining total HMT and *NNMT* expression for *RB1* mutation status as the sole explanatory variable or jointly considered with *NNMT*/HMT expression respectively; *p*-value derived from *t* test. Underlying data for all panels can be found in https://zenodo.org/record/8383542. CCLE, Cancer Cell Line Encyclopedia; GRN, gene regulatory network; HMT, histone methyltransferase; NNMT, nicotinamide N-methyltransferase; Rb, Retinoblastoma; TCGA, The Cancer Genome Atlas; TSS, transcription start site.

To test whether E2F transcription factors also regulated HMT expression in humans, we performed an enrichment analysis for transcription factor binding sites upstream of human HMT genes determined from ENCODE project ChIP-seq experiments for 181 distinct TFs. We observed that E2F1 was the second most significantly enriched TF (Fig 4C).

We identified 256 primary tumours from 10 cancer types in the TCGA that had potentially deleterious mutations in the Retinoblastoma-encoding *RB1* gene in at least 10 samples per cancer type. In all cancer types *RB1*-mutant tumours had higher mean total HMT expression; in a pan-cancer analysis, we observed that this difference was highly significant (Fig 4D). Among 31 single HMTs with expression in all samples, 19 showed a marked up-regulation in *RB1*-mutant tumours, with particularly notable up-regulation of *EZH2*, *DOT1L*, and *NSD2*, while only 5 displayed a clear down-regulation (S7 File).

We also identified cancer cell lines from the CCLE with deleterious Rb mutations. Around half of Rb mutations in CCLE cell lines were found in lung cancer cell lines, particularly small cell lung cancer lines. Total HMT expression was significantly up-regulated in *RB1*-mutant lung cancer cell lines relative to wild-type *RB1* cell lines (Fig 4E; 2-way ANOVA controlling for lung cancer subtype, $p = 0.034$).

To test whether variability in HMT expression across tumours was associated with variable Rb/E2F activity, even if Rb was not mutated, we inferred the activity of 351 transcriptional regulators from expression of target genes in RNA-seq data from thousands of samples across the TCGA and GTEx [47]. In the TCGA, inferred E2F1 activity was significantly (FDR < 0.1) and positively correlated with sample HMT expression in 30/33 cancer types, for example breast cancer (Fig 4F). E2F1 was the ninth transcriptional regulator whose activity most strongly correlated with total HMT expression in a pan-cancer analysis (S6 Table; Spearman's rho = 0.464, 2.56%[ile]). Interestingly, this relationship was much more notable in cancer than healthy tissue (S7 Table). Altogether, these results suggested that Rb activity represses HMT transcription and that this activity is conserved in *C. elegans* and humans.

## In Rb-mutant cancers *NNMT* is down-regulated downstream of HMTs

The preceding results suggested that variation in HMT levels across cancers is associated to variability in Rb and E2F activity and their effects on transcription. *NNMT* expression was significantly reduced in *RB1*-mutant tumours (Fig 4D), consistent with its anticorrelation to HMTs. A key question is how HMT activity might be coordinated with *NNMT*. One possible scenario is that HMTs and *NNMT* might be coordinated by a variable regulator (e.g., Rb) that has opposite effects on the transcription of HMTs and *NNMT*. Alternatively, *NNMT* expression might be regulated to compensate for existing variability in HMT levels, or vice versa (Fig 4G).

We tested this by modelling HMT expression and *NNMT* expression in response either to *RB1* mutation status alone, or in response to the combined effects of *RB1* and either HMTs or

*NNMT*. Inclusion of *NNMT* in the model had no impact on the statistical relationship of HMT expression and *RB1* mutation status (Fig 4H). This is consistent with a direct transcriptional regulation of HMTs by Rb/E2F and demonstrates that expression of HMTs in these tumours is independent of *NNMT* expression. Conversely, including HMT expression when predicting *NNMT* expression abrogates the negative relationship between *NNMT* expression and *RB1* mutation status in tumours (Fig 4H). This is evidence that in *RB1*-mutant cancers, *NNMT* is not directly down-regulated by loss of Rb. Instead, reduced *NNMT* expression is a secondary effect of increased HMT levels.

Altogether, these analyses support the hypothesis that variability in E2F pathway activity drives variation in HMT expression in cancer and that this in turn affects the expression and activity of the *NNMT* methyl sink.

### Artificially reducing SAM/SAH ratio leads to *Nmt* down-regulation in mammalian cells

We wanted to establish whether the transcription of *NNMT* responded to alterations in the SAM/SAH ratios. We identified 2 RNA-seq datasets where the SAH hydrolase *Ahcy* was either knocked out or inhibited pharmacologically in mouse embryonic fibroblasts (MEFs; [48]) and rat hepatic stellate cells [49]. Ahcy loss-of-function causes both SAM and SAH to accumulate and strongly reduces the SAM/SAH ratio [48,50].

We found that reducing Ahcy activity either through *Ahcy* deletion or pharmacological inhibition strongly suppressed *Nmt* expression in both mouse and rat cells (S12A Fig). We also examined how HMT expression responded to artificial reduction of the SAM/SAH ratio. In contrast to the down-regulation of *Nmt*, we observed a moderate up-regulation of HMT expression in response to loss of Ahcy activity in mouse cells and no consistent change in rat cells (S12B Fig).

The high number of replicates in the mouse data gave us sufficient power to search for transcriptional regulators that might control the response of *Nmt* expression to artificial perturbation of the SAM/SAH ratio. Glyr1 was the transcriptional regulator that was most significantly activated by either *Ahcy* deletion or Ahcy inhibition in the mouse dataset (S13A Fig and S8 Table), implying that its activity depends on the SAM/SAH ratio. Prompted by this, we examined the TCGA human primary tumour dataset to test whether GLYR1 might be involved in regulating *NNMT* in human cells. Of 351 transcriptional regulators, GLYR1 activity had both the strongest negative correlation with *NNMT* expression and the strongest positive correlation with total HMT expression (S6 Table). Taking estimated GLYR1 activity into account weakened the statistical relationship between *NNMT* and HMTs in the TCGA more than any other transcriptional regulator (S13B Fig). GLYR1 activity was elevated in *RB1* mutant cancers (S13C and S13D Fig) and taking GLYR1 activity into account abolished the negative statistical relationship between *RB1* mutation and *NNMT* expression (S13E Fig). These results imply a role for GLYR1 in suppressing *NNMT* transcription in response to a low SAM/SAH ratio.

### Discussion

Using data from tens of thousands of human samples, here we demonstrated that HMT expression was strongly anticorrelated to the activity of 2 pathways known to consume excess methyl groups (known as methyl sinks): synthesis of 1MNA by the enzyme NNMT in cancers and production of PC by PEMT in healthy tissues. One possible interpretation of this relationship is that HMTs may also act as a methyl sink. Variation in HMT activity thus might be correlated to the extent to which alternative methyl sink pathways operate: high HMT activity

associated with low NNMT/PEMT activity and vice versa. However, we found no evidence that this variation in HMT activity had an effect on transcription. Below we discuss the implications of these results for understanding the roles of histone posttranslational modifications.

We have shown a strong anticorrelation between HMT levels and NNMT activity in cancer. These results fit with earlier findings that changes in *NNMT* expression could modulate histone methylation [13,16–18]. This was previously argued to be due to a passive effect of NNMT activity on cellular methylation potential via the SAM/SAH ratio. However, we showed that this relationship corresponds to differences in expression of HMTs and *NNMT*. We argue that this is better explained by the hypothesis that elevated histone methylation activity results in a reduced SAM/SAH ratio. We presented evidence that *NNMT* expression changes downstream of HMT expression in Rb-mutant cancers. One possible mechanism for this would be a transcriptional response of *NNMT* to the SAM/SAH ratio. Indeed, we showed that artificially reducing the SAM/SAH ratio by perturbing Ahcy function led to decreased *Nnmt* expression in mouse and rat cells. From these experiments, the transcriptional regulator GLYR1 [51] emerged as a potential link between the SAM/SAH ratio and *NNMT* expression, as we showed GLYR1 likely mediates the relationship between HMT levels and *NNMT* expression in human primary tumours. GLYR1 has been shown to be recruited, via H3K36me3, to the bodies of transcribed genes [52] where it most often promotes gene expression of targets [51]. Since we show that *NNMT* is repressed when GLYR1 activity is high, this suggests it may be an indirect regulator of *NNMT*. In the future, it will be interesting to test whether GLYR1 itself senses the SAM/SAH ratio or whether further intermediate factors are involved.

Our results indicate that coordinated HMT expression is controlled transcriptionally by the activity of the Rb/E2F pathway, such that E2F1 simultaneously activates multiple HMTs associated with both transcriptional activation and repression. It is possible that the coordinated change in HMT activity has a function in buffering the SAM/SAH ratio. However, this would predict that HMT expression should be reduced when the SAM/SAH ratio is reduced. We did not observe this; indeed, HMT activity was actually somewhat increased when the SAM/SAH ratio was artificially reduced by perturbing Ahcy function. Thus, we favour the hypothesis that changes in the SAM/SAH ratio are an important consequence of alterations in HMT activity rather than their primary function. E2F activity increases when cells enter S-phase, which correlates to a demand to introduce methylation marks onto newly synthesised histones [53]. One interesting possibility therefore is that the coordinated transcriptional regulation of multiple HMTs by Rb/E2F is required to maintain epigenetic landscapes through cell division.

The relationship that we have discovered between total HMT activity and the activity of cellular methyl group sinks suggests that maintaining a consistent activity of methyl sink pathways is vital for cellular homeostasis. The importance of this activity might be in buffering cellular methylation potential by converting SAM to SAH to maintain the SAM/SAH ratio. Additionally, SAH is required to support the transsulphuration pathway, which is the cell's only pathway to de novo synthesise cysteine and downstream metabolites (e.g., glutathione). In primary tumours access to cysteine is limited and cells may be forced to rely on transsulphuration [11,54]. However, cultured cells enjoy abundant cysteine supplied in frequently replenished culture medium. In support of this notion, we observed that the HMT/*NNMT* relationship is much stronger in primary tumours than cultured cells. Even so, the relationship is still evident in cancer cell lines and we note that in the CCLE metabolomics data, the first and third metabolites whose levels most strongly correlate positively to total HMT expression are cystathionine and alpha-hydroxybutyrate (Fig 1A), both characteristic markers of transsulphuration [55].

Several studies have placed the SAH hydrolase AHCY in the nucleus associated with chromatin, arguing that it maintains a local environment conducive to histone methylation [56–

59]. However, AHCY-catalysed SAH hydrolysis is reversible and is thermodynamically favoured only when the breakdown products are rapidly metabolised [55]. This hypothesis would therefore predict nuclear activity of either the methionine cycle or the transsulphuration pathway. Given that all of the enzymes required for transsulphuration and glutathione synthesis are annotated as having nuclear localisation (S9 Table) in the Human Protein Atlas [60], the existence of a nuclear transsulphuration pathway fuelled by histone methylation, potentially supplying cysteine for nuclear glutathione synthesis, is an interesting possibility for future exploration.

The histone code hypothesis proposed that specific histone modifications have direct and instructive effects on transcription [2]. However, there are many documented examples where histone-modifying enzymes and the histone marks that they introduce have effects on cellular states that are not due to changes in transcription [61]. Our results provide another such example. We showed that much observed variation in HMT expression is associated with metabolic enzymes with functions far removed from gene regulation. It is conceivable that the metabolic consequences of HMT expression are independent of their catalytic activity in methylation of histone residues. However, we found that the levels of many histone modifications are inversely correlated with *NNMT/PEMT* expression. Thus, the most straightforward implication is that the metabolic consequences of HMT expression are due to their catalytic activity, which consumes SAM. Importantly, this variation does not have any detectable impact on transcription, even as a by-product. These results do not necessarily contradict a role for HMTs in instructing transcriptional regulation. For example, it may be that the variation we observe occurs within a range that does not affect transcription. Alternatively, the changes that occur in histone methylation levels at particular genes may require other changes, such as combinations of histone marks or specific transcription factors, in order to bring about transcriptional responses.

Taken together, our results suggest that histone methylation impacts cellular metabolism, independent of the role of histone methylation marks in regulating transcription. Histone proteins evolved in archaea, where they have a limited role in transcriptional regulation and there is little evidence of posttranslational modifications such as methylation [62]. It is interesting to speculate whether the metabolic consequences of histone posttranslational modifications could predate their more familiar role in transcription.

## Methods

### RNA-seq data

RNA-Seq data were downloaded from the GTEx data portal for GTEx v8. Data were downloaded as raw counts. "Harmonised" (hg38) RNA-seq data were downloaded for TCGA projects using the "TCGAbiolinks" package in "R" as raw counts. CCLE RNA-seq read counts were downloaded from the DepMap download portal in August 2021 (version: DepMap Public 21Q3).

Raw counts were subjected to a median-ratio normalisation (MRN) prior to all analyses. The MRN was performed using the "DESeq2" package in "R" [63]. Normalisations were applied both individually for each tissue or cancer type cohort and across all samples within each database. Normalised pseudocounts were obtained by converting raw counts data to a *DEseq2DataSet* object using the *DESeqDataSetFromMatrix()* function, applying the *estimateSizeFactors()* function to the resulting *dds* object, and then retrieving the normalised pseudocounts with the function *counts()* with *normalised = TRUE*. All correlations presented are based on these MRN-normalised pseudocounts.

For TCGA cancer type analyses, we only considered samples annotated as Primary Tumours, except where we explicitly note otherwise (e.g., adjacent normal tissue samples).

For the CCLE, we excluded samples from Primary Disease types with fewer than 20 cell lines.

We restricted our analyses to GTEx tissues with at least 100 samples or TCGA cancer types with at least 35 Primary Tumour samples.

### Metabolomics and proteomics data

CCLE metabolomics data file "CCLE_metabolomics_20190502.csv" was downloaded from the DepMap download portal. Quantitative proteomics data derived from mass spectrometry for the CCLE was obtained from [15], S2 Table.

Metabolites were manually annotated to KEGG pathways. As metabolites can often be attributed to the function of multiple pathways, we chose appropriate pathways for each metabolite in a heuristic manner aiming to cover a maximum number of metabolites with as few pathways as possible.

### Principal component analysis and hierarchical clustering

Hierarchical clustering of metabolite abundances was performed by running the *hclust()* function in "R" on a distance matrix produced by the *dist()* function on the transposed matrix containing metabolomics data. PCA of metabolite abundances was performed by running the *prcomp()* function in "R" on the transposed matrix containing metabolomics data.

### Correlating metabolites to gene expression

In order to account for biases in cell lines deriving from particular disease types, both metabolite abundances and gene expression were converted to Z-scores for each Primary Disease type prior to correlating metabolite levels to gene expression in the CCLE data. This was done by subtracting the disease type mean abundance from the sample abundance and dividing by the disease type standard deviation for abundance. For RNA-seq pseudocounts, the same approach was taken but using $\log_{10}$-transformed values. These Z-scores were then pooled to perform the correlation analysis. The same approach was taken for correlations of metabolites or gene expression with protein levels measured by proteomics.

For the volcano plot in Fig 1B, the genes correlated to metabolites were limited to a list of 10,275 gold standard genes that are universally expressed across samples (TPM > 5 across all samples in the GTEx data) before calculation of Z-scores. This was done to exclude genes likely to contain samples with 0 values, which would hamper the viable calculation of Z-scores.

Partial correlation analysis was performed on Z-scores as above, using the *pcor()* function in "R."

### Correlation distributions

To ensure equal representation of each tissue or cancer type when combining types across a database, we randomly sampled 100 (GTEx), 36 (TCGA), or 20 (CCLE) from each tissue. Combining raw gene expression data for tissues or cancer types may introduce artifacts even when correcting for average tissue/cancer gene expression, as high-expressing tissues/cancers may still have greater variance in the absolute value of the residuals. To account for this, we ranked the sample gene expression pseudocounts for each gene within the sample chosen for each tissue or cancer type. We then combined the ranks for the chosen samples across tissue types, using the ranks in place of the raw residuals; this gave us 4,800 samples for GTEx, 1,188 samples for TCGA, or 460 samples for the CCLE. The Spearman's correlation was then computed across these aggregated ranks. As the result varies slightly depending on the random

sampling within each tissue, we repeated this process 100 times and plotted the median correlation for each gene.

For analyses within a single tissue type, such a normalisation was not required and we simply correlated the uncorrected pseudocounts for our chosen gene against all others, using all samples available in the cohort.

For gene sets such as HMTs, we added together pseudocounts for each sample for all of the genes before conducting the analysis above.

When computing genome-wide correlations, we did not correct for any potential confounders. Where explicitly noted that values were corrected for confounding variables in the text, we corrected for the following variables in GTEx: age, sex, speed of death, ischemic time, and sequencing batch. For TCGA we corrected for age, race, sex, tumour stage, and sequencing centre.

## Matched cancer and normal samples

Matched primary tumour and adjacent normal tissue samples were identified using TCGA metadata and barcodes. Tissues were identified with at least 30 normal tissue samples. Using the donor portion of the TCGA barcode, matching primary tumour samples were identified. If multiple primary tumour samples matched the adjacent normal tissue sample, one was retained at random and the remainder were discarded. Additionally, normal tissue samples without lacking identifiable primary tumour samples in the expression data were discarded, such that all normal tissue samples had 1 matching primary tumour sample and vice versa.

## Relative reciprocal relationship scores

To calculate relative reciprocal relationship scores in order to compare the strength of gene anticorrelation across different tissue/cancer types, we calculated the genome-wide correlation distribution for both of the interrogated gene (set) pair. We then extracted the genome-wide rank of each of the interrogated gene pair (i.e., a rank of 1 for the most anticorrelated gene), squared these ranks in order to penalise weak reciprocity (i.e., a rank of 1 and 200 in the respective distributions yields a weaker score than ranks of 10 and 10) and added them together to yield the relative reciprocal score.

## Proliferative index

The PI was calculated with the "ProliferativeIndex" package in "R" [41]. Briefly, the entire dataset across all tissues or cancer types was normalised by MRN and variance stabilising transformation using the *varianceStabilizingTransformation()* function of "DESeq2." Following the normalisation, the PI was calculated by applying the *readDataForPI()* function with a randomly selected gene specified in the *modelIDs* argument, then running *calculatePI()* on the resulting object.

## Simulation of correlations among co-regulated genes

We constructed a toy model whereby a theoretical co-regulator positively regulates 40 genes ($A_1$, $A_2$...$A_n$; analogous to HMT genes) and negatively regulates another gene, B. We simulated different concentrations of the co-regulator in 500 different samples, with its influence on $A_{1,2...n}$ and B subject to random noise. We then correlated the simulated concentrations of B to A genes as more A genes are pooled (analogous to our practice of pooling reads for HMT genes). We repeated this simulation 1,000 times.

### Estimations of total immune fraction

The estimates for the immune cell infiltration of TCGA samples using both the TIMER and EPIC RNA-seq deconvolution algorithms were downloaded directly from http://timer.cistrome.org/.

### ENCODE ChIP-seq data processing

We identified publicly available histone methylation ChIP-seq data from adult human patient samples from the ENCODE project for tissues with a strong HMT-*PEMT* correlation in the GTEx data (top 2.5% in both reciprocal correlation distributions) that had at least 3 samples with available RNA-seq data per histone mark. This gave us 19 to 23 samples (depending on the histone mark) from 5 tissues: the esophagus muscularis, the gastroesophageal sphincter, the esophagus squamous epithelium, the sigmoid colon, and the spleen. Data for these samples were available for 4 distinct histone methylation marks: H3K4me3, H3K9me3, H3K27me3, and H3K36me3. For the corresponding ENCODE samples, the following processed data files were downloaded from the ENCODE data portal: ChIP signal fold change over control (as bigwig file) and pseudoreplicated peaks (in bed narrowPeak format). RNA-seq files were downloaded as raw counts for each sample. RNA-seq counts for all samples from all tissues were pooled and MRN-normalised as described above to yield pseudocounts. File names and experiment DOIs are listed in S4 Table.

Fold change over control ChIP-seq signal files downloaded from ENCODE were converted to $log_2$ fold change over control. This was done by using the *bigwigCompare* function of the command line package "deepTools" (v3.5.0) [64] to compare the fold change file against an artificial bigWig file with a flat signal of 1 across all chromosomes, using the argument—*operation log2*.

To select genomic regions in which to model ChIP-seq signal by gene expression, we looked for regions marked by a peak in at least 75% of samples (for H3K9me3, H3K27me3, H3K36me3) or in 100% of samples (H3K4me3, due to greater reproducibility of peak overlaps across samples for this mark). We imported the pseudoreplicated peak files into "R" and used the *countOverlaps()* function of the "GenomicRanges" package against the coordinates of genomic regions of interest to determine the number of peaks that overlapped that region (e.g., a specific promoter) in each sample. We then excluded regions with 0 overlapping peaks in >25% of samples across tissues (or any samples for H3K4me3).

The coordinates of promoters and gene bodies were generated using the "TxDb.Hsapiens.UCSC.hg38.knownGene" package in "R" with the *genes()* and *promoter()* functions (we used the default settings by which the *promoter()* function returns windows from 2,000 bp upstream to 200 bp downstream of each gene's TSS). The coordinates of repeat regions were obtained using the hg38 "rmsk.txt" file downloaded from http://hgdownload.cse.ucsc.edu/goldenpath/hg38/database/.

In order to quantify signal in our chosen genomic regions for each sample, first we exported the genomic coordinates of regions of interest from "R" in bed file format. We then used the *computeMatrix* function of "deepTools" with the sample $log_2$ fold change bigwig file and the genomic regions bed file as inputs. We set the arguments—*binSize* and—*regionBodyLength* to be equal (usually at 100); this results in an output of a single number for average $log_2$ fold change over control ChIP signal for each genomic region.

We went on to model this ChIP signal at each genomic range by expression of chosen gene. Prior to modelling, we filtered out peaks with an average ChIP signal across samples that fell below the level of the input control. From any individual analysis, we first excluded any tissue type that had fewer than 3 samples available for that particular combination of histone mark and tissue, as they could not be effectively modelled.

As multiple ENCODE samples from different tissues often came from the same individual (8 individual donors), we used linear mixed effects models using the "lmer" package in "R" to control for this lack of true sample independence as follows. We first corrected the sample chosen gene expression for tissue and donor of origin by fitting a linear mixed effects model explaining $\log_{10}$ RNA-seq pseudocounts with tissue as a fixed effect and donor as a random effect, extracting the gene expression residual from the model for each sample using the *residuals()* function. We then went on to fit a linear mixed effects model for each genomic region explaining the $\log_2$ fold change over control signal by corrected gene expression residual and tissue as fixed effects and donor as a random effect. From this model, we extracted the model t-value using the *summary()* function as a measure of the explanatory power of gene expression on ChIP signal at that locus and plotted boxplots of all of the t-values for each histone mark and type of genomic region examined.

To generate a null distribution to compare against, we repeated the same modelling procedure with expression of 1,000 random genes from our gold standard set of ubiquitously expressed genes (excluding histone methyltransferases and demethylases). For each genomic region, we then took the mean t-value for the 1,000 random genes as our null distribution that is plotted alongside that of the chosen gene on the boxplot; *p*-values were obtained from a two-tailed paired Wilcoxon test of the observed t-value for our chosen gene for each region against the mean t-value for that region across 1,000 random genes.

To measure the ChIP-seq H3K4me3 peak width, we used the H3K4me3 pseudoreplicated peak narrowPeak bed files downloaded from ENCODE (S4 Table). For each genomic region of interest, we identified overlapping peaks using the *findOverlapPairs()* function of the "GenomicRanges" package in "R." We then calculated the width of these peaks from the start and end coordinates of peak calls from the bed file using the *width()* function. Note we calculated the entire width of the overlapping peak and not only the part which overlapped the region of interest. We then aggregated the peak widths for each genomic region in the case of multiple peaks to provide a single figure for the sum total width of called peaks overlapping that region. We then went on to model peak width as described above for ChIP-seq signal.

When modelling or correlating gene expression from promoters, we restricted the analysis to genes with expression detected in every sample.

## NCI60 ChIP-seq data processing

To probe the relationship of histone methylation levels in chromatin to NNMT in cancer, we used the NCI60 [65], a panel of 60 cancer cell lines with associated RNA-seq, microarray, proteomics, and ChIP-seq data.

NCI60 ChIP-seq data were taken from [65]. We downloaded raw sequencing reads from the NIH's Sequence Read Archive, using the command line tool "SRAtoolkit" (SRA identifier numbers found in S5 Table). We then aligned the reads to the human genome (hg38) using the "bowtie2" package [66]. Each experiment was available as 2 replicates, in addition to an input control sample. Some sequencing replicates consisted of single-reads, while others were paired-end experiments. In order for all analyses to be comparable, we aligned only the forward reads from paired-end experiments. The output from "bowtie2" was saved to a.sam file and converted to a.bam file using the "samtools" package (version 1.16.1). bam files were then sorted by query name using the *sort* function from "samtools."

We used the "Genrich" package (https://github.com/jsh58/Genrich) to call ChIP-seq peaks on the sorted.bam files, as "Genrich" can take 2 replicate experimental files, in addition to a control file, and perform an integrated peak call relative to the control. For all marks (H3K4me3, H3K9me3, and H4K20me3), we used the following settings, corresponding to

"broad" peaks: -q (max FDR) 0.1, -g (max distance between significant sites) 400. Additionally, for H3K4me3 we used the default settings, corresponding to "narrow" peaks. We then imported these peak call files into "R" and used the *reduce()* function of the "GenomicRanges" package to combine the peaks called under broad and narrow settings for H3K4me3.

For each experimental replicate, we used the bamCompare function of "deepTools" with—*operation log2* to return a signal file in bigWig format for $log_2$ fold change over control. We identified genomic regions of interest (promoters, gene bodies, and repeats) as described above for ENCODE, limiting to genomic sites with a called peak in >2/3 of samples. We then ran the *computeMatrix* function of "deepTools" with the sample $log_2$ fold change bigwig file and the genomic regions.bed file as inputs, with equal bin size and body length as described above, to return a single figure for average $log_2$ fold change for each region. At this point, we took the average of the 2 replicates and carried that forward into our modelling approach.

We downloaded raw RNA sequencing reads using "SRAtoolkit" (SRA identifier numbers in S5 Table) and aligned them to the human genome (hg38) using "bowtie2" using default settings. We calculated read counts per gene by counting reads overlapping exons using the *summarizeOverlaps* function of the "GenomicAlignments" package in "R." Read counts were then MRN-normalised before use in modelling. We downloaded NNMT / *NNMT* SWATH mass-spectrometry proteomics values and 5-microarray gene expression Z-scores for NCI60 cell lines from CellMinerCDB (https://discover.nci.nih.gov/rsconnect/cellminercdb/).

For each genomic region, we modelled the average $log_2$ fold change over control signal using a generalised linear model with the *glm()* function of the "stats" package in "R." In the model, we included the expression of our gene/protein of interest (sample RNA-seq pseudo-counts, protein level, or mRNA Z-score), in addition to the cell line tissue of origin and an interaction term between tissue and expression. We extracted and plotted the model t-values for expression as described above.

## Mouse embryonic fibroblast and rat liver RNA-seq data processing

For MEFs, we downloaded raw reads with "SRAtoolkit" as described above from GEO Accession GSE126851. We used "bowtie2" to align reads to the *Mus musculus* genome (build GRCm39) and used "featureCounts" from the "Subread" package with the GRCm39 gtf annotation downloaded from ensemble.org to generate gene-level read counts. These were then normalised with DESEq2 prior to plotting or TR activity estimation. For rat liver samples treated with DZNep, gene level raw counts were downloaded from the Supplementary Material of GEO Accession GSE121736 and treated as above. *M. musculus* and *Rattus norvegicus* HMT genes were taken to be orthologues of human HMTs and are found in S1 Table.

## Bayesian iterative reweighting analysis of multi-mapping ChIP-seq reads in NCI60

Both ChIP-seq signal and RNA-seq expression levels from repetitive elements are difficult to quantify accurately due to ambiguous multi-mapping of sequencing reads to highly similar genomic regions. These reads are typically discarded in data processing pipelines (as in the ENCODE pipeline). However, for the NCI60, we used a Bayesian iterative reweighting approach ("SmartMap") to apportion multi-mapping reads to individual genomic loci, providing more accurate estimates of ChIP-seq signal at repetitive elements [67]. We performed this analysis for H3K9me3 and H4K20me3.

The 2 replicates in the NCI60 ChIP-seq data, as well as being paired and unpaired, have different read lengths. Additionally, the input controls are unpaired reads with shorter read lengths (150 bp). In the case of multi-mapping reads, greater read length was likely to affect

the likelihood of unique mapping and so affect the validity of comparisons to the input control. As such, we only made use of a single replicate for each histone mark and cell line, namely the unpaired replicate with shorter read lengths that matched the input control. We adapted the *SmartMapPrep* script from the "SmartMap" package to process raw single-end reads, down-loaded using "SRAtoolkit" as described above, for treatment and input controls, before using the *SmartMap* function for a single reweighting iteration as default and as recommended by the authors. The output from "SmartMap" is a bedgraph file. We converted the bedgraph files to bigwig files with the UCSC "bedGraphToBigWig" utility. Log$_2$ fold signal over input control was found using deepTools *bigwigCompare* with the default "log2"—operation choice.

For peak calling, we used "deepTools" *bigwigCompare* with the—*operation subtract* setting to remove the signal from the appropriate input control track from each track. We then converted these bigWig files back to bedgraph files with the UCSC *bigWigToBedGraph* utility. We used MACS3 (v3.0.0) [68] for peak calling, as it can call peaks from a bedgraph file using the *bdgbroadcall* function.

As above we analysed individual repetitive elements that were marked by a peak in at least 40 of 60 cell lines. Repetitive elements were identified from the "rmsk.txt" file as described above, with the exception of HERVs, which were taken from the annotation included in the Telescope package (see below). Custom bed files were created with the elements to be analysed and signal was quantified across the entire element using "deepTools" *computeMatrix* as described above.

## Estimation of HERV expression in the NCI60

We also used a separate Bayesian reweighting approach ("Telescope") to estimate locus-specific expression estimates from a set of HERVs [69]. We aligned raw RNA-sequencing reads (downloaded with "SRAtoolkit" as above) with "bowtie2" with options—*very-sensitive-local* and *-k 100* (allowing up to 100 alignments per read), as recommended by the "Telescope" package authors. The resulting bam files were processed in "Telescope" using the *telescope assign* function call. The HERVs annotation file "HERV_rmsk.hg38.v2.gtf" was downloaded from the "telescope_annotation_db" repository on GitHub (https://github.com/mlbendall/telescope_annotation_db). We analysed individual HERVs marked by ChIP peaks (identified using "SmartMap") in 40 of 60 cell lines and which had expression detected in at least 30 of 60 cell lines. When using HERV expression as a response variable in a linear model, we used negative binomial generalised linear model (with the *glm.nb()* function)due to typical overdispersion of the data.

## Correlations within histone methyltransferase genes

To probe co-expression of HMT genes, we first corrected the expression values for confounding variables as described above. Additionally for the GTEx pan-tissue analysis, we corrected for donor ID as a random variable within a linear mixed effects model, to account for the fact that when comparing across tissues multiple samples can originate from the same individual donor. Corrected residuals were rank-percentile transformed within each tissue or cancer type, before 100 (GTEx) or 36 (TCGA) samples were chosen from each and combined before Spearman's correlations among the rank-percentile transformed values were computed across the grouped samples. The process of sampling was repeated 100 times and the median Spearman correlation from the 100 iterations was taken for plotting. We excluded very lowly expressed HMT genes from plots by filtering according to a geometric mean expression across all samples of at least 100 pseudocounts; this accounts for slightly different numbers of HMT genes in the GTEx/TCGA plots shown in S10 Fig.

Correlations of HMT genes to random genes were computed as above with 100 random genes from our gold standard set of ubiquitously expressed genes for each HMT. Values computed were then pooled.

Network plots were prepared from pan-cancer or pan-tissue correlation matrices by first filtering out edges with correlations of magnitude less than 0.2. Node size was based on its degree and edges were weighted by the square of the magnitude of the correlation. Network analysis was performed in "Gephi" (version 0.10.1), with the following visualisation properties: ForceAtlas2 layout, edge weight range 0.1 to 2.0 and attraction 30 for TCGA and 10 for GTEx.

## De novo transcription factor binding motif search

We used "MEME" (version 5.5.1) in discriminative mode to find motif occurrences that were enriched in the Cel-HE cluster promoters relative to the remaining HMT promoters. We used 1,000 bp upstream of the TSS as our promoter sequences. The previously published EFL-2 motif [70] was obtained from CisBP [71] version 1.02; motif identifier M0675_1.02.

## CeNDR *C. elegans* data and processing

We downloaded RNA-seq raw counts and TPM-normalised values for *C. elegans* strains from the *C. elegans* Natural Diversity Resource (CeNDR; [72]) from Gene expression Omnibus, accession number GSE186719. TPM values were used only for estimating sample ages using the "RAPToR" tool (see below). We MRN-normalised the raw counts and used these counts for all other analyses. The raw counts were transcript-level counts; these were collapsed down to gene-level counts prior to all analyses.

We used the "RAPToR" package in "R" [73] to infer the age of the samples according to the author's instructions. We used the *Cel_YA_2* reference series from the "wormRef" package. To obtain age-corrected residuals for gene expression, we fitted a spline with 6 degrees of freedom using the *smooth.spline()* function of the "stats" package in "R" to predict $\log_{10}$ pseudocounts from inferred age, taking the residuals from the spline with the *residuals()* function.

*C. elegans* HMT genes were selected according to their gene descriptions on WormBase (version WS287). The list of *C. elegans* HMTs can be found in S1 Table. The vast majority of strains were represented by 3 independent RNA-seq samples. We took the mean of the age-corrected residuals for each strain to plot scatterplots and compute HMT correlations for the heatmap. *C. elegans* orthologues of human HE cluster genes were determined using OrthoList 2 [74]; a gene was annotated as an orthologue if the orthology relationship was present in at least 3 of the 6 databases compiled in OrthoList 2.

## *C. elegans lin-35* mutants and ChIP-seq data

To identify RNA-seq datasets from *lin-35* mutants, we searched the Gene Expression Omnibus for *lin-35* and found 2 studies; [45] (GEO accession GSE62833) for *lin-35* mutant or wild-type L3 larvae and [46] (GEO accession GSE155190) for L1 larvae. We downloaded the raw data and aligned it to the *C. elegans* genome (version WS276) using "bowtie2" with default settings, before obtaining gene level counts using *summarizeOverlaps()* from "GenomicRanges" as described above for NCI60 and then MRN-normalising the resulting counts with "DESeq2" as described above. In both studies total HMT counts were increased in *lin-35* mutants on average by 12% to 15%. To assess statistical significance, we analysed the 2 studies together, performing a 2-way ANOVA for total HMT counts with genotype and developmental stage as explanatory variables.

LIN-35 ChIP-seq data was obtained from [44], S1 Table. We used "GenomicRanges" in "R" to determine the distance from reported significant LIN-35 peaks to the TSS of HMT genes.

For "other genes," we excluded all genes which have 0 expression in any sample; we then randomly sampled 1,000 of the remaining approximately 11,000 genes.

## Rb-mutant cancers in the TCGA and CCLE

Mutation calls for TCGA samples were downloaded as MAF files using the "TCGABiolinks" package in "R" [75]. We identified samples with a reported mutation in *RB1* which was either a missense mutation, nonsense mutation, frame-shifting insertion, in-frame deletion, or a frame-shift deletion. Ten cancer types had at least 10 *RB1*-mutant samples with available RNA-seq data; BLCA, BRCA, CESC, COAD, HNSC, LIHC, LUSC, LUAD, SARC, and UCEC. We assumed samples had wild-type *RB1* if they had mutations called in other genes in the MAF files but none called in *RB1*.

We rank-percentile transformed all samples in these cancer types (*RB1* mutant and wild type together) and then sampled 10 *RB1*-mutant and 10 wild-type cancers from each cancer type, taking the median rank-percentile of each random sample. We repeated this sampling process 1,000 times and plotted the medians from these samples for each group, comparing the medians with a *t* test.

To model HMT or *NNMT* expression by *RB1* mutation status and/or counterpart expression, we rank-percentile transformed HMT and *NNMT* expression by cancer type after correction for confounding variables. We then performed pan-cancer sampling of *RB1*-mutant or wild-type cancers as above, combining all samples and fitting a linear model with HMT/*NNMT* expression as response variable and either *RB1*-mutation status alone as an explanatory variable or together with counterpart expression. We then extracted the t-values from the various linear models for the statistical association of *RB1*-mutation to either HMT or NNMT expression.

For the CCLE, we downloaded the mutation calls using the *depmap_mutationCalls()* function of the "depmap" package in "R." We then filtered *RB1* mutations by whether they were called as deleterious or not. We found 90/1,236 cell lines had deleterious *RB1* mutations, of which 43 were annotated as lung cancer cell lines, 32 specifically small cell lung cancer. As small cell lung cancers had a higher expression of HMTs than other lung cancer subtypes, we performed a 2-way ANOVA with genotype and lung cancer subtype as explanatory variables.

## Transcriptional regulator activity estimation

To estimate transcriptional regulator (TR) activity in GTEx and TCGA samples, we used the "decoupleR" package in "R" [47] to infer TR activity from RNA-seq samples. "decoupleR" requires a gene regulatory network (GRN) to use as a basis for TR activity inference; we used the "dorothea" package in "R" [76] previously developed by the same authors. The "dorothea" package includes 2 different human GRNs; 1 general and 1 for cancers. We used the general GRN for estimating GTEx sample TR activity and the cancer GRN for TCGA sample TR activity. In order to increase our confidence in the estimates, we excluded the lowest confidence TR-target interactions that were the result only of in silico predictions. We then excluded any TRs that had fewer than 10 target genes remaining by which to infer their activity and also excluded any that were in our list of HMTs. This left us with 351 transcriptional regulators. Before running "decoupleR," we also weighted targets by the confidence of the interaction, converting the confidence reported by "dorothea" (letters A–D after low-confidence interactions had been eliminated) into an integer value [1–4] and using its inverse as an interaction weight.

For estimating TR activity in MEFs, we used the *dorothea_mm* GRN from the "dorothea" package, which is based on the human GRN. To find TRs whose targets were enriched among

differentially expressed genes in the Ahcy loss-of-function datasets, we performed a differential expression analysis with "DESeq2" in "R." We used the *results()* function to obtain the test statistics for all genes. We then used these test statistics for all genes as input for "decoupleR" using the mlm method to detect enriched TRs. In order to plot Glyr1 activity in S13 Fig, we estimated TR activities separately for each sample with "decoupleR" using counts normalised with "DESEq2" as input.

## ENCODE transcription factor binding enrichment

We downloaded the ENCODE transcription factor binding site profiles from the "Harmonizome" database web portal [77]. For the genes bound by each transcription factor, we performed a Fisher's exact test for enrichment of HMT genes among the bound genes. Odds ratios and p values are extracted from the Fisher's exact test; *p*-values reported in the volcano plot are raw and uncorrected.

## Supporting information

**S1 Fig. 1MNA is an outlier when clustering metabolite levels across CCLE cell lines.**
(TIF)

**S2 Fig. Total histone methyltransferase expression is strongly anticorrelated with the activity of NNMT in cancers (related to Fig 1).**
(TIF)

**S3 Fig. Infiltration of specific immune cell types into primary tumours correlates positively with *NNMT* expression but does not strongly confound the HMT-*NNMT* relationship.**
(TIF)

**S4 Fig. In melanoma *NNMT* expression is strongly anticorrelated with the histone methyltransferase-encoding *gene SETDB1*, a known driver of melanoma.**
(TIF)

**S5 Fig. Relationship of other groups of methyltransferases to *NNMT* in cancers.**
(TIF)

**S6 Fig. *NNMT* and HMT expression and relationship in cancer vs. normal samples.**
(TIF)

**S7 Fig. Relationship of other groups of methyltransferases to *PEMT* in healthy tissues.**
(TIF)

**S8 Fig. *PEMT* expression anticorrelate globally with levels of specific histone marks genome-wide in healthy tissues (related to Fig 3).**
(TIF)

**S9 Fig. *NNMT* expression anticorrelate globally with levels of specific histone marks genome-wide in cancer cell lines (related to Fig 3).**
(TIF)

**S10 Fig. Highly expressed histone methyltransferase genes are co-expressed.**
(TIF)

**S11 Fig. Histone methyltransferases are co-expressed and correlate with *NNMT/PEMT* orthologues/analogues in *C. elegans*.**
(TIF)

**S12 Fig. Alteration of SAM/SAH ratio transcriptionally regulates Nnmt and HMTs.**
(TIF)

**S13 Fig. The transcriptional regulator GLYR1 may mediate NNMT down-regulation.**
(TIF)

**S1 Table. Methyltransferase gene sets used in this study.**
(XLSX)

**S2 Table. CCLE metabolite-RNA-seq correlations for HMTs and NNMT.**
(XLSX)

**S3 Table. CCLE metabolite-proteomics correlations for HMTs and NNMT.**
(XLSX)

**S4 Table. ENCODE ChIP-seq and RNA-seq files.**
(XLSX)

**S5 Table. NCI60 ChIP-seq and RNA-seq files.**
(XLSX)

**S6 Table. HMT correlations to transcription factor activity in TCGA.**
(XLSX)

**S7 Table. HMT correlations to transcription factor activity in GTEx.**
(XLSX)

**S8 Table. Differential TF activity in Ahcy loss of function experiments.**
(XLSX)

**S9 Table. Subcellular localisations of transsulphuration and glutathione synthesis genes.**
(XLSX)

**S1 File. CCLE *NNMT*-HMT correlation all cancers full labelled plots.**
(PDF)

**S2 File. TCGA *NNMT*-HMT correlation all cancers full labelled plots.**
(PDF)

**S3 File. GTEx *NNMT*-HMT correlation all tissues full labelled plots.**
(PDF)

**S4 File. GTEx *PEMT*-HMT correlation all tissues full labelled plots.**
(PDF)

**S5 File. GTEx individual tissue HMT correlation matrices.**
(PDF)

**S6 File. TCGA individual tissue HMT correlation matrices.**
(PDF)

**S7 File. Individual HMT levels in Rb-mutant tumours in TCGA.**
(PDF)

## Acknowledgments

We thank the members of the EpiEvo group and other colleagues at the Department of Biochemistry, University of Oxford for helpful discussion and comments on the project.

## Author Contributions

**Conceptualization:** Marcos Francisco Perez, Peter Sarkies.

**Data curation:** Marcos Francisco Perez.

**Formal analysis:** Marcos Francisco Perez, Peter Sarkies.

**Funding acquisition:** Peter Sarkies.

**Supervision:** Peter Sarkies.

**Visualization:** Marcos Francisco Perez, Peter Sarkies.

**Writing – original draft:** Marcos Francisco Perez, Peter Sarkies.

**Writing – review & editing:** Marcos Francisco Perez, Peter Sarkies.

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
