## [Editor Report · Decision Letter 0]

4 May 2023

Dear Peter, 

Thank you for submitting your manuscript entitled "Histone methylation has a direct metabolic role in human cells" for consideration as a Research Article by PLOS Biology.

Your manuscript has now been evaluated by the PLOS Biology editorial staff as well as by an academic editor with relevant expertise and I am writing to let you know that we would like to send your submission out for external peer review. However, we would like to consider the manuscript as a Short Report and we will decide upon the reviewers' responses if that format is the best fit. Thus, please select that type of article from the drop down menu when you submit the metadata (see below), but you can submit the paper as it is.

Before we can send your manuscript to reviewers, we need you to complete your submission by providing the metadata that is required for full assessment. To this end, please login to Editorial Manager where you will find the paper in the 'Submissions Needing Revisions' folder on your homepage. Please click 'Revise Submission' from the Action Links and complete all additional questions in the submission questionnaire.

Once your full submission is complete, your paper will undergo a series of checks in preparation for peer review. After your manuscript has passed the checks it will be sent out for review. To provide the metadata for your submission, please Login to Editorial Manager (https://www.editorialmanager.com/pbiology) within two working days, i.e. by May 08 2023 11:59PM.

Kind regards,

Ines

--

Ines Alvarez-Garcia, PhD

Senior Editor

PLOS Biology

---

## [Decision Letter · Decision Letter 1]

28 Jun 2023

Dear Peter,

Thank you for your patience while your manuscript entitled "Histone methylation has a direct metabolic role in human cells" was peer-reviewed at PLOS Biology as a Short Report. Please also accept again my apologies for the delay in providing you with our decision. The manuscript has now been evaluated by the PLOS Biology editors, an Academic Editor with relevant expertise, and by three independent reviewers. 

The reviews are attached below. As you will see, the reviewers find the conclusions novel and potentially very interesting, but they also raise several concerns regarding the evidence provided to support the main claims, which they find too correlative. As a result, the reviewers think that some of the claims are overstated and that they would need to either be supported by further evidence or toned down, including the title.

Also, please note that our Short Reports only have four figures, thus one of the figures will have to be converted into a Supplementary figure during the revision.

In light of the reviews and the Academic Editor assessment, we would like to invite you to revise the work to thoroughly address the reviewers' reports and we expect you to add further data to better support the conclusions, but we would leave up to you which experiments you would like to perform and which ones you would like to address in the text by toning down some of the claims.

Given the extent of revision needed, we cannot make a decision about publication until we have seen the revised manuscript and your response to the reviewers' comments. Your revised manuscript is likely to be sent for further evaluation by all or a subset of the reviewers.

**IMPORTANT - SUBMITTING YOUR REVISION**

3. Resubmission Checklist

a) *PLOS Data Policy* - IMPORTANT - please read

b) *Published Peer Review*

Sincerely,

Ines

--

Ines Alvarez-Garcia, PhD

Senior Editor

PLOS Biology

Reviewers' comments

Rev. 1:

In this manuscript, the authors first performed correlation analyses between the expression level of histone methyltransferases (HMTs) and either metabolite levels or expression of other genes. They found 1) negative correlations between HMTs and the metabolite 1-methylnicotinamide (1MNA), a product of methylation of nicotinamide by the enzyme NNMT, in cancer cell lines, 2) negative correlations between HMTs and NNMT in both cancer cell lines and tumors, and 3) negative correlations between HMTs and PEMT, another enzyme catalyzing a methylation reaction, in normal tissues. In the second part of the manuscript, the authors focused on the positive correlations among HMTs and through bioinformatic analysis identified E2F and Rb as transcriptional regulators of HMTs.

The manuscript shows clear and intriguing associations between HMTs and levels of 1MNA methyl sink enzymes. And the identification of two new transcriptional regulators of HMTs is another major result. My main concern however is that the main claim of the manuscript remains a hypothesis and is beyond the data.

As stated in the title, the main claim is that "Histone methylation has a direct metabolic role in human cells". To make this claim, the author would need to show that it is the activity of HMTs that cause the changes in 1MNA and the expression of NNMT or PEMT. The correlations found do not support any causal relation. The only piece of data for supporting a causal relation (that HMTs are upstream of the methyl pool and other methyl sink pathways) is the analysis of gene expression data of Rb, HMTs, and NNMT (Fig. 5G-H). This is however very indirect evidence as it requires the establishment of Rb as the regulator of HMTs in the first place. Furthermore, the effect size shown in Fig. H is small, reducing the strength of this piece of evidence.

I understand that direct manipulations of the HMT activities would require major experimental effort and may be beyond the scope of the study. But the authors should tune down their claim or focus on the E2F and Rb part of their study.

Rev. 2:

This manuscript presents evidence that histones function as methyl sinks in both normal and cancer cells. The authors have carefully analyzed numerous publicly available datasets, noting an inverse correlation between the expression of histone methyltransferases (HMTs) and nicotinamide N-methyltransferase (NNMT) in cancer cells, as well as phosphatidylethanolamine N-methyltransferase (PEMT) in normal tissues. Both of these enzymes, previously shown to consume S-adenosylmethionine (SAM) to generate 1-methylnicotinamide (1MNA) and phosphatidylcholine (PC), respectively, also produce S-adenosylhomocysteine (SAH) as a byproduct. Given this anticorrelation, the established influence of NNMT or PEMT on the regulation of the SAM/SAH ratio, and an extensive examination of histone modifications, the authors argue that histone methylation represents an alternative mechanism for adjusting the SAM/SAH ratio. Their further analysis suggests that changes in histone methylation are likely independent of gene expression. Lastly, they provide interesting evidence indicating that E2F and Rb probably control the coordinated expression of HMTs as a group.

The manuscript is compelling and makes ingenious use of published datasets. The findings are intriguing, their data analysis is thorough and sound, and their conclusions are well substantiated by the provided evidence. The paper is very well-written and was a pleasure to read. I recommend it for publication, subject to minor revisions. I have the following questions and suggestions.

Given the alternative sinks for methyl groups, could the authors provide insight into the factors/conditions that dictate the preferential use of one sink over another? For example, why is NNMT predominantly used in some cancer cells/conditions, while HMTs are employed in others?

The transsulphuration pathway facilitates the conversion of SAM to SAH, homocysteine, and subsequently cysteine, which can be synthesized into downstream products such as glutathione (GSH). Do the authors notice any correlations between HMTs, NNMT, or PEMT with enzymes involved in cysteine biosynthesis? Could a limitation in cysteine supply influence the preference for one pathway over another, perhaps due to flux or other physiological parameters optimizing SAM to SAH conversion?

Lower levels of histone methylation and acetylation have been shown to predict poor outcome in multiple cancer types. Do the authors observe concordant changes in histone acetylation and methylation, especially in the analyses of the cancer tissues? And do these changes correlate with outcome? For instance, do tumors with high NNMT and low histone methylation (or low HMT expression) fare worse than those with low NNMT and high histone methylation? If the authors have access to such datasets, integrating them could bolster the clinical relevance of their findings to cancer prognosis and potentially offer insight into how cells select among various methyl sinks.

Lastly, the authors use the term "sink" to suggest that the deposition of methyl groups on histones might be terminal as far as the methyl groups are concerned. However, are there known mechanisms to recover methyl groups from histones,1MNA or PC molecules? If so, would it not be more appropriate to use the term "storage" instead?

Rev. 3:

In this study titled 'Histone methylation has a direct metabolic role in human cells', the authors propose that methyl transferase enzymes control the role of different methyl sinks, and thereby regulate the methylation status in cells. The authors find anti-correlations between histone methyl transferases and other methyl sinks such as NNMT and PEMT enzymes in cancers and healthy tissues. They also uncover a possible role for the Retinoblastoma (Rb) protein in regulating the expression of HMT and NNMT. The authors thereby propose that this mode of regulation of methyl sinks can modulate SAM homeostasis, and thereby the cellular metabolic state.

The strength of this manuscript are the correlations between HMT and the other methyl transferases-NNMT and PEMT in cancers and healthy tissues and uncover the possible mode of regulation of HMT expression. However, the role of these in regulating the cellular SAM pools or metabolic state is not clear, and the claim that this is involved "directly in core, conserved pathways of cellular metabolism" is an overstatement. There are multiple really interesting correlations from this study, and the central hypothesis is very nice, but the authors try to connect these with cellular metabolism without sufficient evidence and this overshadows their actual findings. The following are the concerns with this study.

Major concerns:

1) The title of the study "Histone methylation has a direct metabolic role in human cells" is very broad, and the study does not justify this title. The authors have done extensive analysis to understand how the expression of different methyl transferases correlate. Even though the results can be extended to understand how methyl sinks in the cells are regulated, in the absence of specific analyses to address this question, the title is an exaggeration. Similarly, the abstract and discussion also makes strong claims about how this impacts metabolism, with insufficient data.

2) It is not clear whether and how HMT expression varies between healthy and normal cells. In fig 1A, HMT expression is negatively corelated with lactate, which is often increased in cancers. Is HMT levels generally downregulated in cancers? If a comparison is made between healthy and cancer cells, how does the expression of different methyl transferases change and what will be the correlation among them?

3) One of the bigger concerns regarding this study is that it does not address how the changes in the expression of HMT, PEMT and NNMT actually regulate the SAM or SAH levels. Do SAM, SAH (or methionine) levels change in Rb mutant tumors where HMT expression is upregulated? Or does the decrease in NNMT, which compensates for the HMT increase, stabilize the SAM and SAH pools? In the absence of evidence from such experiments, the role of HMT and other methyl transferases in regulating methylation pool is severely overstated.

4) Correspondingly, is there any correlative data to suggest that changes in HMT, NNMT or PEMT occur when there are defects methionine or SAM metabolism? If there is a defect or decrease in SAM synthesis (and therefore the methylation source), would the expression of these methyl transferase enzymes change correspondingly to compensate for these? This would strengthen the correlative argument that the changes in these enzymes have a direct role in regulation methylation pool/ SAM metabolism. In the absence of data related to the same, the claim that "our results indicate histone methylation participates directly in core, conserved pathways of cellular metabolism" is not supported. I'd encourage the authors to think hard about their statements, and what makes very strong arguments, vs which ones are weaker.

5) How is the expression of PEMT correlated with other methylation/methyltransferase enzymes, especially NNMT in cancer cells vs healthy tissues? There is enough evidence that phosphatidylcholine metabolism is differentially regulated in cancers. Will changes in PEMT expression, and its correlation with other methyl transferase expression tell us more about how methylation states are controlled in cancers?

6) Is there any information regarding the localization of methyl transferases, their correlation between their expression? For example, is the correlation between methyl transferases present in the same cellular compartment better compared to those in other compartments? Is this involved in regulating SAM pools in specific compartments such as nucleus, cytosol and mitochondria? If there is some data in this space, it might allow the authors to make a much stronger correlative claim, related to metabolic state and the function/activity of methyltransferases.

Minor concerns:

1) Fig 1E should be placed before the figure which introduces NNMT. The current placement breaks the flow from 1D to 1F.

---

## [Decision Letter · Decision Letter 2]

2 Sep 2023

Dear Peter,

Thank you for your patience while we considered your revised manuscript entitled "Histone methyltransferase activity affects metabolism in human cells" for publication as a Short Report at PLOS Biology. This revised version of your manuscript has been evaluated by the PLOS Biology editors, the Academic Editor and one of the original reviewers.

Based on the review, we are likely to accept this manuscript for publication, provided you satisfactorily address the data and other policy-related requests stated below.

In addition, we would like you to consider a suggestion to improve the title:

"Histone methyltransferase activity affects metabolism in human cells independently of transcription regulation"

We expect to receive your revised manuscript within two weeks. 

*Published Peer Review History*

*Press*

Sincerely,

Ines

--

Ines Alvarez-Garcia, PhD

Senior Editor

PLOS Biology

DATA POLICY:

Fig. 1A-C, E-I; Fig. 2A-G, I-M; Fig. 3A-D; Fig. 4B-F, H; Fig. S2A-E, G-J; Fig. S3A-F; Fig. S4A-E; Fig. S5; Fig. S6A-D; Fig. S7; Fig. S8A-F; Fig. S9A-E; Fig. S10A, B; Fig. S11A-E; Fig. S12A, B and Fig. S13A-E

CODE POLICY

Per journal policy, as the code that you have generated is important to support the conclusions of your manuscript, we require that you make it available without restrictions upon publication. You mention in the manuscript that the code is available via the GitHub page for this project. Please send us the ID of the project and DOIs and make it publicly available. Please ensure that the code is sufficiently well documented and reusable, and that your Data Statement in the Editorial Manager submission system accurately describes where your code can be found.

Reviewers' comments

Rev. 3:

This is a much improved revision, and makes much clearer, more nuanced statements with direction (histone methyltransferase  metabolism).

---

## [Editor Report · Decision Letter 3]

27 Sep 2023

Dear Dr Sarkies,

Thank you for the submission of your revised Short Report entitled Histone methyltransferase activity affects metabolism in human cells independently of transcriptional regulation" for publication in PLOS Biology. On behalf of my colleagues and the Academic Editor, Jason Locasale, I am delighted to let you know that we can in principle accept your manuscript for publication, provided you address any remaining formatting and reporting issues. These will be detailed in an email you should receive within 2-3 business days from our colleagues in the journal operations team; no action is required from you until then. Please note that we will not be able to formally accept your manuscript and schedule it for publication until you have completed any requested changes.

PRESS

Sincerely, 

Ines

--

Ines Alvarez-Garcia, PhD

Senior Editor

PLOS Biology
